# Cellular acidosis triggers human MondoA transcriptional activity by driving mitochondrial ATP production

Blake R Wilde, Zhizhou Ye, Tian-Yeh Lim, Donald E Ayer*

Department of Oncological Sciences, Huntsman Cancer Institute, University of Utah, Salt Lake City, United States

**Abstract** Human MondoA requires glucose as well as other modulatory signals to function in transcription. One such signal is acidosis, which increases MondoA activity and also drives a protective gene signature in breast cancer. How low pH controls MondoA transcriptional activity is unknown. We found that low pH medium increases mitochondrial ATP (mtATP), which is subsequently exported from the mitochondrial matrix. Mitochondria-bound hexokinase transfers a phosphate from mtATP to cytoplasmic glucose to generate glucose-6-phosphate (G6P), which is an established MondoA activator. The outer mitochondrial membrane localization of MondoA suggests that it is positioned to coordinate the adaptive transcriptional response to a cell's most abundant energy sources, cytoplasmic glucose and mtATP. In response to acidosis, MondoA shows preferential binding to just two targets, TXNIP and its paralog ARRDC4. Because these transcriptional targets are suppressors of glucose uptake, we propose that MondoA is critical for restoring metabolic homeostasis in response to high energy charge.
DOI: https://doi.org/10.7554/eLife.40199.001

## Introduction

Glucose is a major source of carbons for the production of ATP and biosynthetic intermediates. Dys-regulation of glucose uptake and metabolism underlies many diseases including cancer and diabetes (*Petersen et al., 2017*; *Hay, 2016*). Thus, it is important to understand the precise molecular mechanisms that regulate glucose homeostasis in normal and pathological settings.

The paralogous transcription factors MondoA and ChREBP (MondoB) are sentinel regulators of glucose-induced transcription and their activity is highly, if not entirely, dependent on glucose (*Stoltzman et al., 2008*; *Richards et al., 2017*; *Peterson et al., 2010*; *Stoltzman et al., 2011*; *Ma et al., 2005*). Work by our lab and others has established glucose-6-phosphate (G6P) as a key regulatory signal that drives Mondo transcriptional activity (*Stoltzman et al., 2008*; *Li et al., 2010*). Other hexose-6-phosphates, fructose-2,6-bisphosphate, and xylulose-5-phosphate are also thought to drive Mondo-dependent transcription, yet the molecular mechanisms are not well-defined (*Kabashima et al., 2003*; *Petrie et al., 2013*; *Stoltzman et al., 2011*).

MondoA controls the glucose-dependent expression of thioredoxin-interacting protein (TXNIP), which has a number of critical cellular functions (*Anderson, 2016*; *Shalev, 2014*; *O'Shea and Ayer, 2013*). The best characterized among these is as a potent suppressor of glucose uptake (*Stoltzman et al., 2008*; *Wu et al., 2013*; *Hui et al., 2008*). Thus, MondoA and TXNIP – the MondoA/TXNIP axis – constitute a negative feedback loop that maintains cellular glucose homeostasis. High TXNIP is anti-correlated with glucose uptake in human tumors and is a predictor of better overall survival in cancer patients, establishing the MondoA/TXNIP axis as an important prognostic factor in cancer (*Lim et al., 2012*; *Chen et al., 2010*; *Shen et al., 2015*).

*For correspondence:
don.ayer@hci.utah.edu

Competing interests: The authors declare that no competing interests exist.

MondoA shuttles from the outer mitochondrial membrane (OMM) to the nucleus where it drives TXNIP expression (*Billin et al., 2000*; *Sans et al., 2006*; *Stoltzman et al., 2008*). TXNIP is among a handful of characterized MondoA targets, yet the full scope of the direct MondoA-transcriptome has not been reported. In addition to being regulated by glucose, a functional electron transport chain (ETC) is also required for MondoA-dependent transcription (*Yu et al., 2010*; *Han and Ayer, 2013*), yet the ETC-derived signal remains unknown. It is also unclear how glycolytic and mitochondrial signals converge to regulate MondoA transcriptional activity. Nevertheless, because MondoA responds to both glycolysis and mitochondrial respiration, MondoA may function as a master sensor of cellular energy charge.

TXNIP expression is driven by a number of cellular stresses, including serum starvation, lactic acidosis/low pH, ultraviolet and gamma irradiation, endoplasmic-reticulum stress and microgravity (*Elgort et al., 2010*; *Chen et al., 2010*; *Junn et al., 2000*; *Versari et al., 2013*; *Oslowski et al., 2012*). However, little is known about how TXNIP expression is regulated by this diverse collection of signals. TXNIP expression is highly dependent on MondoA and glucose, suggesting that at least some of these stresses impact MondoA activity and/or the availability of glucose-derived metabolites.

Intracellular acidification is a metabolic stress intrinsic to proliferating cells that results from increased glycolytic flux and consequent lactate production. Cancer cells initiate an adaptive response to intracellular acidification to restore physiological pH that includes export of lactate, slowing of glycolysis and restriction of glucose uptake (*Webb et al., 2011*; *Gunnink et al., 2014*). pH-regulation of glycolytic flux and proton transport have been well-studied (*Webb et al., 2011*), and our previous work suggests a role for the MondoA/TXNIP axis in normalizing cellular pH. For example, lactic acidosis triggers MondoA-dependent TXNIP expression and decreased glucose uptake (*Chen et al., 2010*). This suppression of glucose uptake requires both MondoA and TXNIP, yet the mechanisms by which lactic acidosis activates MondoA transcriptional activity was not investigated.

Here we show that acidic pH drives MondoA transcriptional activity by increasing mitochondrial ATP (mtATP) synthesis. mtATP is used by mitochondria-bound hexokinase to generate G6P from cytoplasmic glucose, which subsequently drives MondoA nuclear accumulation and transcriptional activity. These results suggest a critical role for the MondoA in coordinating the transcriptional and metabolic response to the cell's principal energy sources, glucose and mtATP, and in maintaining energy homeostasis in response to nutrient hyper-abundance.

## Results

### Low pH medium drives MondoA transcriptional activity

We previously showed that lactic acidosis drives MondoA localization to the TXNIP promoter (*Chen et al., 2010*), raising the possibility that lowering intracellular pH increases MondoA transcriptional activity. Supporting this hypothesis, TXNIP mRNA expression is inversely correlated with three critical regulators of proton export that function to increase intracellular pH (*Webb et al., 2011*): monocarboxylate transporter (MCT) 1 in lung cancer, MCT4 in breast cancer and sodium-hydrogen antiporter 1 (NHE1) in brain cancer (*Figure 1—figure supplement 1A*). In contrast, TXNIP expression positively correlated with an acidosis gene-signature in breast cancer (*Figure 1—figure supplement 1B*). TXNIP expression was also anti-correlated with MCTs and NHE1 in non-transformed tissues (*Figure 1—figure supplement 1C and D*), demonstrating that pH-dependent regulation of MondoA activity at the TXNIP is not restricted to cancer cells. These data suggest that intracellular pH per se, rather than a lactic acidosis-dependent signaling event, controls MondoA transcriptional activity.

To dissect how acidosis controls MondoA transcriptional activity, we treated cells with Hank's balanced salt solution (HBSS): HBSS has minimal pH-buffering capacity and in 5% $CO_2$ has an acidic pH of ~6.5. HBSS treatment of murine embryonic fibroblasts (MEFs) increased TXNIP mRNA and protein expression, and decreased glucose uptake (*Figure 1A and B*, *Figure 1—figure supplement 2A*). HBSS is weakly buffered due to a low level of sodium bicarbonate (0.35 g/L). Supplementing HBSS with sodium bicarbonate to 3.7 g/L raised the pH to 7.5 and prevented TXNIP induction (*Figure 1C*, *Figure 1—figure supplement 2B*). Conversely, decreasing sodium bicarbonate in DMEM to 0.37 g/

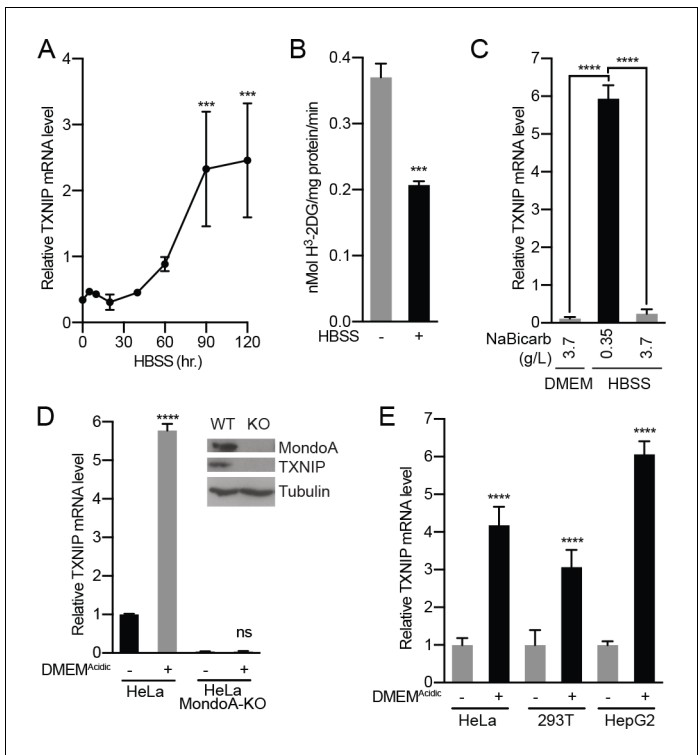

**Figure 1.** Acidosis drives MondoA transcriptional activity. (**A**) TXNIP mRNA levels in murine embryonic fibroblasts (MEFs) following treatment with HBSS for the indicated times. (**B**) Glucose uptake was determined by quantifying the rate of $^3$H-2-deoxyglucose uptake in MEFs following a 4 hr treatment with HBSS. (**C**) TXNIP mRNA levels from MEFs treated for 4 hr with DMEM, HBSS and HBSS supplemented with sodium bicarbonate to the amount in DMEM (3.7 g/L). (**D**) CRISPR/Cas9 was used to disrupt expression of MondoA in HeLa cells. Amounts of the indicated proteins in HeLa and HeLa:MondoA-KO cells were determined using immunoblotting. Consistent with our previous findings TXNIP expression was highly dependent on MondoA. TXNIP mRNA levels from HeLa and HeLa:MondoA-KO cells following a 4 hr treatment with DMEM$^{Acidic}$. (**E**) TXNIP mRNA levels in HEK-293T, HeLa and HepG2 cells following 4 hr treatments with DMEM$^{Acidic}$. In A, C, D and E, TXNIP mRNA levels were determined by reverse transcriptase-quantitative PCR (RT-qPCR). ***p<0.001; ****p<0.0001; ns – not significant.
DOI: https://doi.org/10.7554/eLife.40199.002

The following figure supplements are available for figure 1:

**Figure supplement 1.** TXNIP expression correlates with genes that regulate intracellular pH.
DOI: https://doi.org/10.7554/eLife.40199.003

**Figure supplement 2.** Acidosis drives MondoA transcriptional activity.
DOI: https://doi.org/10.7554/eLife.40199.004

**Figure supplement 3.** Acidosis drives MondoA transcriptional activity.
DOI: https://doi.org/10.7554/eLife.40199.005

L (DMEM$^{Acidic}$) decreased the pH to ~6.5 and induced TXNIP expression (*Figure 1—figure supplement 2C*). To confirm that TXNIP induction is mediated by pH rather than sodium bicarbonate, we increased the pH of HBSS and DMEM$^{Acidic}$ to 7.4 and varied the sodium bicarbonate concentration. This slightly alkaline pH blocked TXNIP induction regardless of sodium bicarbonate concentration. (*Figure 1—figure supplement 2C*), confirming that low pH rather than low sodium bicarbonate is primarily responsible for HBSS- and DMEM$^{Acidic}$-driven MondoA transcriptional activity.

MondoA is necessary and sufficient for TXNIP induction (*Stoltzman et al., 2011*; *Stoltzman et al., 2008*). Consistent with this, TXNIP was induced in HeLa cells treated with DMEM$^{Acidic}$ but not in HeLa cells where we used CRISPR/Cas9 editing to disrupt MondoA expression (*Figure 1D*). Likewise, HBSS failed to increase TXNIP expression in MondoA$^{-/-}$ MEFs but reconstituting MondoA null MEFs with human MondoA rescued TXNIP induction (*Figure 1—figure*

supplement 3A). DMEM$^{Acidic}$ induced TXNIP expression in cell lines of different lineages: HeLa, HepG2 and 293 T cells (Figure 1E), indicating a conserved regulatory mechanism.

We next investigated how acidosis influences MondoA transcriptional activity. MondoA(I766P), which does not interact with Mlx (Stoltzman et al., 2008), did not rescue TXNIP induction in MondoA null MEFs (Figure 1—figure supplement 3A), which is consistent with several previous findings showing that MondoA requires Mlx as an obligate partner (Stoltzman et al., 2011; Peterson et al., 2010; Minn et al., 2005; Stoltzman et al., 2008). HBSS also induced the activity from a TXNIP-promoter luciferase reporter, but not when the MondoA:Mlx Carbohydrate Response Element (ChoRE) binding site sequence was mutated (Figure 1—figure supplement 3B and C). Finally, HBSS treatment led to increased MondoA occupancy at the TXNIP promoter (Figure 1—figure supplement 3D). Together these data establish that acidosis drives MondoA nuclear accumulation, promoter binding and transcriptional activity.

## MondoA is dependent upon mitochondrial ATP

Previous reports show that low pH growth medium drives intracellular acidification (Adams et al., 2006; Wahl et al., 2000). To determine the site of action of low pH on MondoA transcriptional activity, we initially used compartment-selective ionophores to alter proton concentrations. Monensin, and the mitochondrial ionophore FCCP, which drive cytosolic alkalization and dissipation of the mitochondrial proton gradient, respectively, completely block HBSS-induced TXNIP expression (Figure 2A–B). By contrast, chloroquine, which disrupts acidification of endosomes/lysosomes, had no effect on TXNIP induction (Figure 2A). Together these results suggest that cytosolic and/or mitochondrial proton gradients, but not pH-dependent changes in the endosome/lysosome, are critical for the activation of the MondoA/TXNIP axis.

Cytosolic and mitochondrial protons contribute to ETC activity and previous inhibitor studies demonstrate that MondoA requires a functional ETC for transcriptional activity (Yu et al., 2010; Han and Ayer, 2013). Therefore, we first performed a series of genetic tests to investigate how the ETC contributes to acidosis-driven MondoA activity. We used 143Bρ$^0$ osteosarcoma cells, which lack mitochondrial DNA and are respiration deficient (King and Attardi, 1989). TXNIP was induced in parental 143B cells treated with DMEM$^{Acidic}$ (Figure 2C), yet induction was blunted in 143Bρ$^0$ cells (Figure 2D). TXNIP induction was rescued in 143Bρ$^0$ cells that had been repopulated with wild type mitochondria (143Bρ$^0$:WT-cybrid cells; Figure 2E). These genetic experiments confirm the previous inhibitor studies that implicated a functional ETC in MondoA transcriptional activity. Surprisingly, treatment of cells with the ETC complex I inhibitor metformin did not block TXNIP induction in response to DMEM$^{Acidic}$ suggesting that the ETC requirement may be downstream of complex I (Figure 2B).

Given the predominant role of the ETC in ATP synthesis, we next investigated whether mitochondrial ATP (mtATP) synthesis is required to trigger MondoA transcriptional activity. We used 143Bρ$^0$:ΔATP6/ΔATP8 cybrid cells. These cells have a point mutation in mtDNA that disrupts expression of both ATP6 and ATP8, which are required components of the $F_0F_1$-ATPase (ATP synthase, Figure 2F) and consequently synthesize less mitochondrial ATP (Boominathan et al., 2016; Jonckheere et al., 2008). Low pH-driven TXNIP expression was reduced in these cells, yet was partially rescued by expression of nuclear-encoded, but mitochondrially-targeted ATP6 and ATP8 (Figure 2G) (Boominathan et al., 2016). These results suggest that mtATP synthesis is necessary for low pH to induce MondoA transcriptional activity. Complementing these two genetic experiments, the ATP synthase inhibitor oligomycin completely blocked TXNIP induction in response to DMEM$^{Acidic}$ (Figure 2B).

## Acidosis drives the synthesis of mitochondrial ATP

The experiments above suggest a critical role for mtATP in driving MondoA transcriptional activity and raise the possibility that acidosis increases MondoA transcriptional activity by driving mtATP synthesis. Because the outer mitochondrial membrane is freely permeable to protons (Cooper, 2000), we hypothesized that acidosis generates an ectopic proton gradient across the inner mitochondrial membrane and the mitochondrial matrix that subsequently drives mtATP synthesis. We tested this hypothesis in several ways. First, DMEM$^{Acidic}$ treatment shifted intracellular pH from 7.2 to 6.5 (Figure 3A). Second, the drop in intracellular pH was accompanied by an increase in

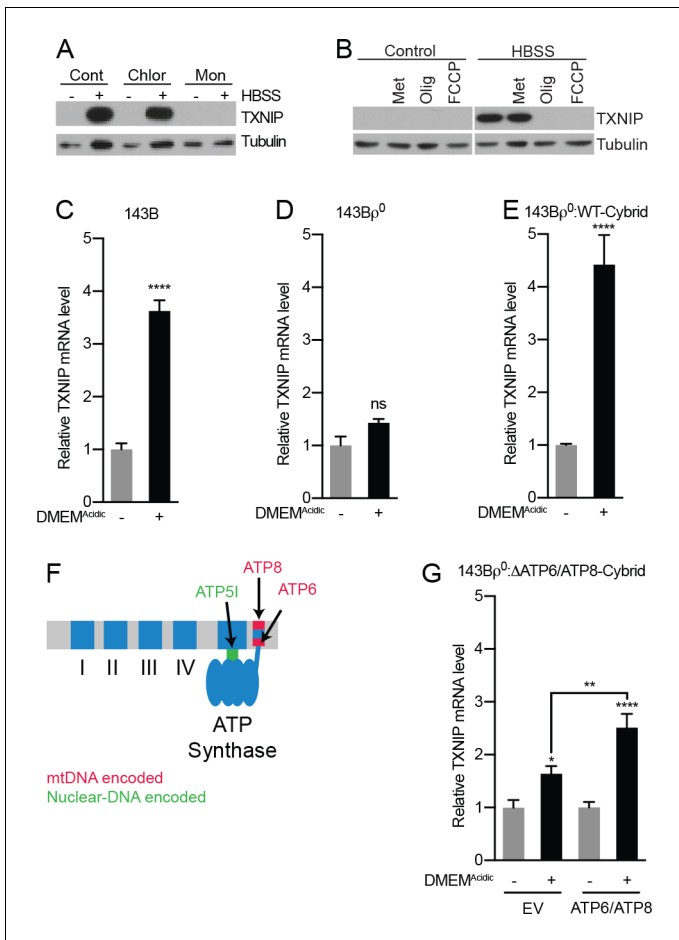

**Figure 2.** Acidosis-driven MondoA transcriptional activity depends on electron transport. (**A and B**) TXNIP protein levels were determine by immunoblotting following a 4 hr HBSS treatment of MEFs in the presence of the indicated inhibitors. Cont; control, Chlor; Chloroquine (25 µM), Mon; monensin (5 µM), Met; metformin (1 mM), FCCP; carbonilcyanide $p$-triflouromethoxyphenylhydrazone (1 µM), Oligo; oligomycin (1 µM). (**C–E**), TXNIP mRNA levels were determine using RT-qPCR following 4 hr treatments with DMEM$^{Acidic}$: (**C**) 143B osteosarcoma cells, (**D**) 143Bρ$^0$ cells, and (**E**) 143Bρ$^0$:WT-Cybrid cells complemented with wild type mitochondria. (**F**) Schematic depicting nuclear- and mitochondrial-DNA encoded components of the ETC. (**G**) TXNIP mRNA levels following 4 hr DMEM$^{Acidic}$ treatments of 143Bρ$^0$:ΔATP6/ATP8-Cybrid cells expressing empty vector or nuclear encoded, mitochondrial-targeted ATP6 and ATP8. *$p<0.05$; **$p<0.01$; ****$p<0.0001$; ns – not significant.

DOI: https://doi.org/10.7554/eLife.40199.006

mitochondrial membrane potential (*Figure 3B*). Third, DMEM$^{Acidic}$ treatment increased in total cellular ATP levels (*Figure 3C*). Thus, treating cells with low pH medium increases total cellular ATP levels. Others have shown that acidic medium can suppress mTORC1 activity (*Walton et al., 2018*), which under some circumstances can result in an ~10% increase in total ATP that stems from a blockade of mTORC1-dependent translation (*Zheng et al., 2016*). However, even with an extend time course of DMEM$^{Acidic}$ treatment, we did not observe significant suppression of mTORC1 activity as assayed by levels of phosphorylated-S6 (*Figure 3—figure supplement 1*). This finding suggests that DMEM$^{Acidic}$ increases TXNIP expression by a mechanism that is largely independent of mTORC1 and its control of ATP pools.

DMEM$^{Acidic}$ increased total cellular ATP content (*Figure 3C*), however, metabolite levels in whole cell lysates can be vastly different from those observed in specific organelles (*Abu-Remaileh et al., 2017*; *Chen et al., 2016*). Therefore, we sought to determine how DMEM$^{Acidic}$ affects ATP levels in different cellular compartments. To accomplish this, we used a fluorescence resonance energy transfer (FRET)-based ATP biosensor that consists of cp173-Venus fused to mseCFP via an ATP-

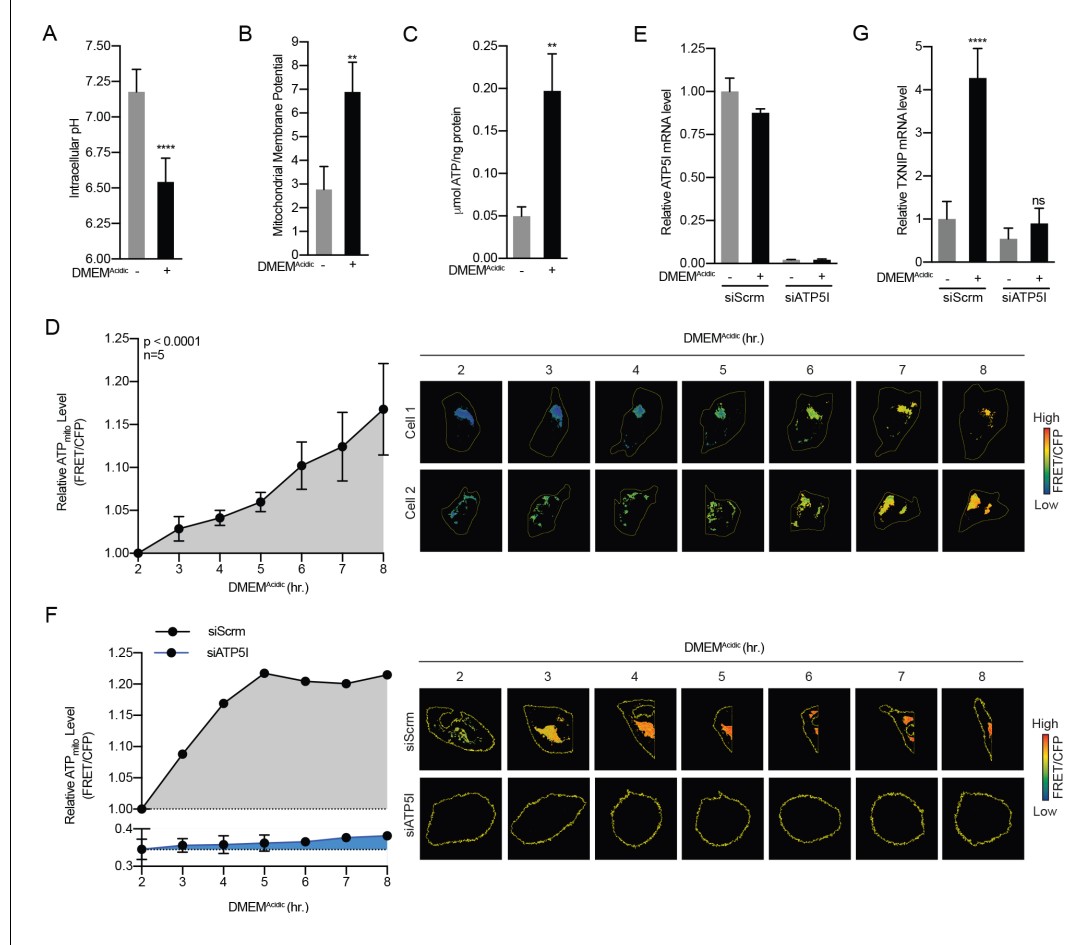

**Figure 3.** Acidosis drives the synthesis of mitochondrial ATP. (**A**) Intracellular pH of HeLa cells was determined using BCECF-AM staining after a 4 hr treatment with DMEM$^{Acidic}$. (**B**) Mitochondrial membrane potential of HeLa cells was determined by JC1 staining after a 1 hr treatment with DMEM$^{Acidic}$. (**C**) Total cellular ATP levels in a lysate prepared from HeLa cells was determined using a luciferase-based assay after a 4 hr treatment with DMEM$^{Acidic}$. (**D**) Mit-ATEAM, which is a mitochondrial-targeted ATP-biosensor, was used to determine the relative level of mtATP in HeLa cells treated with DMEM$^{Acidic}$ for the indicated times. Widefield microscopy was used to capture images in the FRET and CFP channels. Images were then used to analyze FRET and CFP signals in mitochondria in individual cells. FRET signal was normalized using CFP. (**E**) ATP5I mRNA levels in HeLa cells that expressed scrambled (siScrm, n = 1) or ATP5I-specific siRNA (siATP5I, n = 2) following a 4 hr DMEM$^{Acidic}$ treatment. (**F**) Mit-ATEAM was used to determine relative mtATP levels in HeLa cells transfected with siScrm or siATP5I and treated with DMEM$^{Acidic}$ for the indicated times. (**G**) TXNIP mRNA levels following a 4 hr DMEM$^{Acidic}$ treatment of HeLa cells that had been transfected siScrm or siATP5I. In (**E and G**), ATP5I and TXNIP mRNA levels were measured using RT-qPCR. **p<0.01; ****p<0.0001; ns – not significant.

DOI: https://doi.org/10.7554/eLife.40199.007

The following figure supplements are available for figure 3:

**Figure supplement 1.** DMEM$^{Acidic}$ does not inhibit mTORC1 activity Immunoblotting was used to measure the levels of the indicated proteins following DMEM$^{Acidic}$ treatment of HeLa cells for the indicated times.

DOI: https://doi.org/10.7554/eLife.40199.008

**Figure supplement 2.** Acidosis drives synthesis of mitochondrial ATP.

DOI: https://doi.org/10.7554/eLife.40199.009

binding domain and generates FRET signal upon ATP binding (*Imamura et al., 2009*). ATEAM measures cytoplasmic ATP pools, whereas Mit-ATEAM, which is targeted to the inner-mitochondrial matrix, measures mtATP pools (*Figure 3—figure supplement 2A–B*). HeLa cells treated with DMEM$^{Acidic}$ showed increased FRET from Mit-ATEAM but not from ATEAM indicating that acidosis drives an increase in mtATP but not in cytosolic ATP (*Figure 3D*, *Figure 3—figure supplement 2C–E*). As expected, constructs with mutations in the ATP-binding linker that prevent ATP binding showed no FRET signal (*Figure 3—figure supplement 2C*). DMEM$^{Acidic}$ treatment increased total ATP pools,

yet we only observed an increase in mtATP and not in cytosolic ATP in this assay (*Figure 3—figure supplement 2D and E*). This finding suggests that DMEM[Acidic] treatment increases total cellular ATP levels primarily by increasing mtATP synthesis.

We next sought to determine whether the accumulation of mtATP resulted from increased synthesis or decreased mitochondrial export. We blunted expression of ATP5I, an essential component of the ATP synthase, using siRNA-mediated knockdown (*Figure 3E*). Consistent with our working model, ATP5I knockdown not only decreased the steady state level of mtATP, but also the DMEM[Acidic]-driven increase in mtATP (*Figure 3F*). Furthermore, ATP5I knockdown prevented TXNIP induction in response to DMEM[Acidic] treatment (*Figure 3G*). Together these data suggest that low pH medium drives mtATP production through ATP synthase and that mtATP synthesis is required for DMEM[Acidic] to drive MondoA transcriptional activity.

## MondoA senses G6P produced by mitochondrial-hexokinase

How does MondoA sense mtATP? MondoA, Mlx and hexokinase 2 (HK2) are all resident at the outer mitochondrial membrane (*Figure 4A–B*) (*Robey and Hay, 2006*; *Sans et al., 2006*). Mitochondria-bound HK2 has preferential access to mtATP as it is exported from the mitochondria (*Wilson, 2003*) and it transfers the terminal phosphate from ATP to cytoplasmic glucose to generate G6P. Because G6P is a known activator of MondoA transcriptional activity, we speculated that DMEM[Acidic]-induced mtATP synthesis drives G6P synthesis resulting in a stimulation of MondoA transcriptional activity (*Figure 4B*). We have tested this model in several ways. First, we determined how acidosis alters steady-state metabolite levels. Consistent with reports showing that acidosis leads to increased mitochondrial metabolism (*Lamonte et al., 2013*; *Chen et al., 2008*; *Dietl et al., 2010*), DMEM[Acidic]

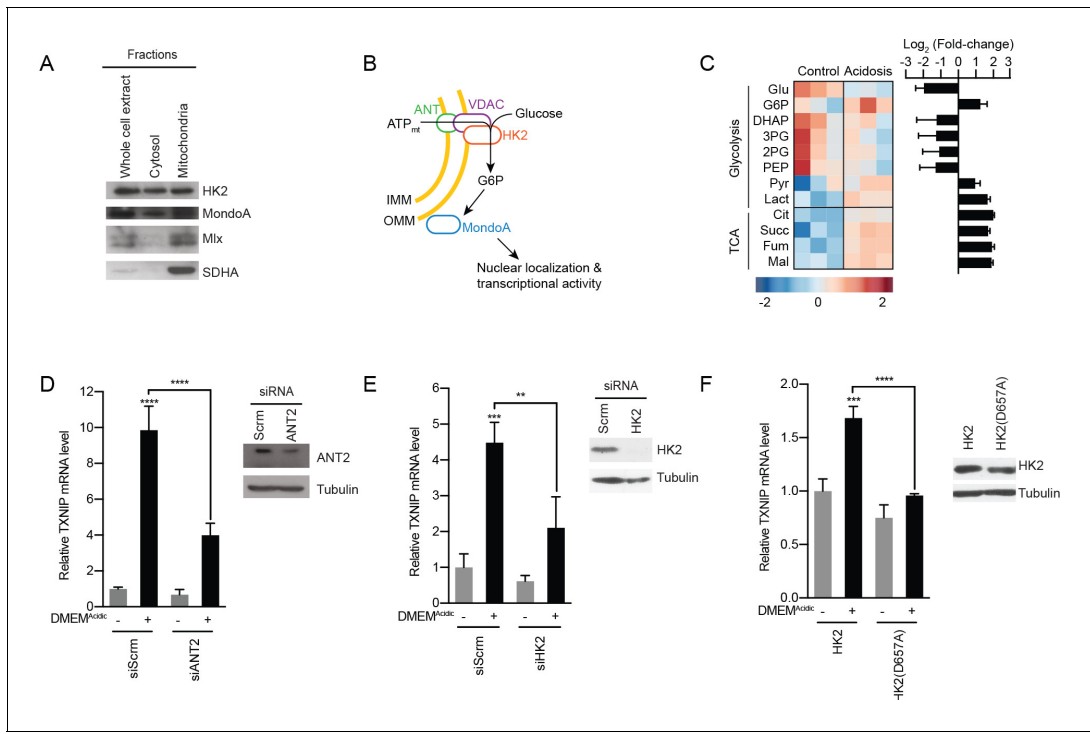

**Figure 4.** MondoA is activated by G6P produced by hexokinase utilization of mtATP. (**A**) Cellular fractionation of BJ-Tert cells indicating mitochondrial localization of HK2, MondoA and Mlx. Succinate dehydrogenase A (SDHA) serves as a control for the mitochondria fraction. (**B**) Schematic illustrating how mtATP could contribute to MondoA transcriptional activity. As mtATP is exported from the mitochondria, it is used by as a substrate by mitochondria-bound HK2 to produce G6P, resulting in MondoA activation. (**C**) Heatmap and log$_2$ fold-changes of glycolytic and TCA metabolites measured using GC-MS following a 4 hr treatment of HeLa cells with DMEM[Acidic]. TXNIP mRNA levels, as measured by RT-qPCR, following 4 hr DMEM[Acidic] treatments of HeLa cells and expressing pools of four siRNAs against (**D**) ANT2 and (**E**) HK2, or (**F**) expressing HK2 and HK2(D657A). Immunoblots validate knockdown of the indicated proteins. **p<0.01; ***p<0.001; ****p<0.0001; ns – not significant.
DOI: https://doi.org/10.7554/eLife.40199.010

drove an increase in TCA cycle intermediates (*Figure 4C*). Most glycolytic intermediates were decreased in response to DMEM$^{Acidic}$; however, G6P levels increased 3-fold (*Figure 4C*).

Second, we tested the contribution of the channel, comprised of the adenine-nucleotide transporter (ANT) in the inner-mitochondrial membrane and voltage-dependent anion channel (VDAC) in the outer-mitochondrial membrane, that exports mtATP from the mitochondria matrix. Consistent with our working model, which states that mtATP must be exported from the matrix to drive MondoA transcriptional activity, siRNA-mediated knockdown of ANT2 prevented TXNIP induction in response to low pH medium (*Figure 4D*). This finding suggests that mtATP functions outside the mitochondria to trigger MondoA transcriptional activity, rather than by an indirect signaling-based mechanism.

Third, we used several approaches to test the contribution of HK2 to low pH-driven MondoA activity. Consistent with our previous reports (*Stoltzman et al., 2008*), siRNA pools against HK2 blocked TXNIP induction in response to low pH treatment (*Figure 4E*), demonstrating a requirement for HK2. Further, overexpression of HK2(D657A), which lacks emzymatic activity (*Arora et al., 1991*), blocked the induction of TXNIP in response to DMEM$^{Acidic}$ (*Figure 4F*), supporting the notion that HK2 activity and the formation of G6P is critical for the induction of MondoA transcriptional activity.

Fourth, we tested the contribution of mitochondria-localized HK2 to low pH-driven MondoA transcriptional activity. HK2 localizes to the outer-mitochondrial membrane via interactions with VDAC (*Wilson, 2003*). Ectopic expression of mVDAC(E72Q), a mutant of the mouse orthologue of VDAC1, which prevents the interaction between VDAC and HK2 (*Abu-Hamad et al., 2008*; *Zaid et al., 2005*), blocked the mitochondrial localization of HK2 as expected and completely blocked TXNIP induction in response to DMEM$^{Acidic}$ (*Figure 5A*). We complemented these loss-of-function experiments with a gain-of-function approach designed to determine whether mitochondrial localization of hexokinase was sufficient for low pH-induced MondoA transcriptional activity. To accomplish this goal, we artificially tethered HK2 to the mitochondria by fusing VDAC1 to the first 10 β-strands of GFP (mVDAC1-GFP(1-10)) and by fusing HK2 to the last β-strand of GFP (HK2-GFP(11)). When co-expressed, the β-strands of GFP self-assemble (*Kamiyama et al., 2016*), linking mVDAC1 and HK2 (*Figure 5B*). Expression of mVDAC1(E72Q)-GFP(1-10), which does not interact with HK2, blocked TXNIP induction (*Figure 5B–C*). However, co-expression of mVDAC1(E72Q)-GFP(1-10) and HK2-GFP (11) rescued HK2 mitochondrial localization and TXNIP induction (*Figure 5B–C*). Together these data show that mitochondrial-localized HK2 is both necessary and sufficient for low pH medium to drive MondoA activity.

Finally, we determined how the different regulatory components that comprise our model affect MondoA nuclear localization. While the majority of MondoA resides in the cytosol (*Figure 5D*), DMEM$^{Acidic}$ treatment drove an increase in MondoA nuclear localization in cells treated with an siRNA control (*Figure 5D*); however, MondoA nuclear accumulation was blunted in cells treated with siRNA pools against ATP5I, ANT2 or HK2 (*Figure 5D*). In combination with the above experiments, these findings suggest that mtATP synthesis and the subsequent production of G6P by OMM-bound HK2 drive MondoA nuclear accumulation, occupation of the TXNIP promoter and activation of TXNIP transcription in response to low pH medium.

## MondoA is required for the transcriptional response to acidosis

To determine the contribution of MondoA to acidosis-driven gene expression we conducted RNA-sequencing on mRNA from HeLa and HeLa:MondoA-KO cells treated with DMEM$^{Acidic}$ for 4 hr. Using a 1.5-fold cut off and an adjusted p-value of $\leq$0.01, we identified 617 differentially regulated genes in HeLa cells treated with DMEM$^{Acidic}$. Of these, 227 were not regulated in HeLa:MondoA-KO cells, suggesting that MondoA contributes to nearly 37% of the acidosis-driven transcriptional response. We next used regression analysis to look for genes that are affected by both DMEM$^{Acidic}$ treatment and genotype. Loss of MondoA prevented the induction/suppression of several acidosis-regulated genes; however, the expression only two genes, TXNIP and one of its paralogues, ARRDC4, were entirely dependent on MondoA (*Figure 6A*).

We next performed pathway analysis on genes differentially regulated in HeLa and HeLa:MondoA-KO cells treated with DMEM$^{Acidic}$. Consistent with the results above, TXNIP and ARRDC4 were the most highly MondoA-dependent genes, with log$_2$(fold-change) decrease of 7.9 and 5.1, respectively (*Figure 6B*). We identified 157 other differentially regulated genes in HeLa:MondoA-KO cells (adjusted p-value$\leq$1E-10). Pathways that were upregulated in HeLa:MondoA-KO cells were enriched

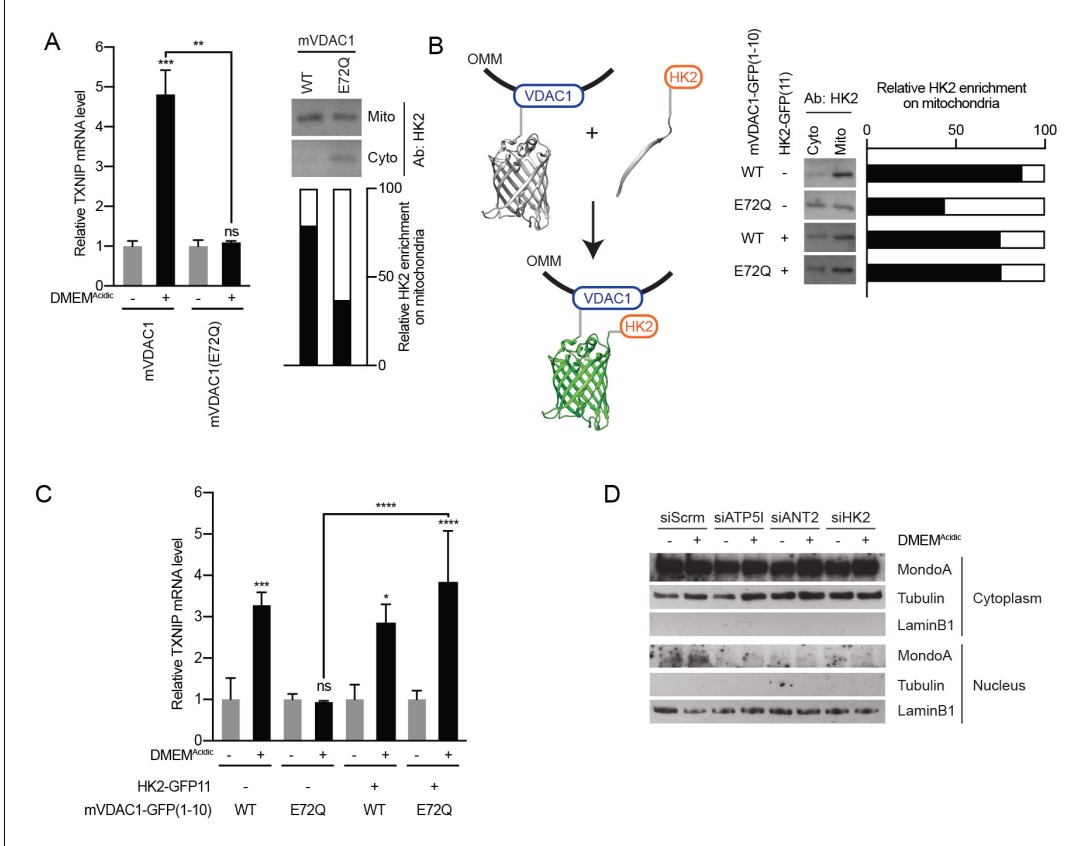

**Figure 5.** MondoA senses G6P produced by mitochondrial-bound hexokinase. (**A**) TXNIP mRNA levels in BJ-Tert cells expressing mVDAC1-GFP and mVDAC1(E72Q)-GFP and treated for 4 hr with DMEM$^{Acidic}$. HK2 localization was also analyzed by cellular fractionation and densitometry was used to quantify the relative amount of HK2 localized to the mitochondria. HK2 was increasingly enriched in the cytoplasmic fraction in cells expressing mVDAC1(E72Q), suggesting that it had been displaced from mitochondria. (**B**) Schematic depicting the use of GFP(1-10) and GFP(11) to artificially tether mVDAC1 and HK2. HK2 localization was also analyzed by cellular fractionation and densitometry was used to quantify the relative amount of HK2 present in the mitochondrial fraction. (**C**) TXNIP mRNA levels in BJ-Tert cells treated for 4 hr with DMEM$^{Acidic}$ and expressing mVDAC1-GFP, mVDAC1 (E72Q)-GFP and HK2-GFP(11) in the indicated combinations. (**D**) MondoA nuclear localization was determined by cellular fractionation of HeLa cells treated for 4 hr with DMEM$^{Acidic}$ and transfected with siScrm (siRNA control), siATP5I, siANT2 or siHK2. Tubulin and LaminB1 served as controls for cytoplasmic and nuclear fractions, respectively. In (**A and C**), TXNIP mRNA levels were determined by RT-qPCR. *p<0.05; **p<0.01; ***p<0.001; ****p<0.0001; ns – not significant.

DOI: https://doi.org/10.7554/eLife.40199.011

for fatty acid metabolism, sterol biosynthesis, ion homeostasis, ROS metabolism, and pyridine metabolism pathways, whereas cell death and proliferation pathways were downregulated (*Figure 6B*, *Figure 6—source data 1*). Further, we conducted Gene Set Enrichment Analysis (GSEA) on HeLa and HeLa:MondoA-KO cells treated with DMEM$^{Acidic}$ using all pathways in the Molecular Signatures Database. We identified 588 gene sets that were enriched with a nominal p-value<0.001 (*Figure 6—source data 2*). Leading-edge analysis highlighted extracellular matrix remodeling, electron transport chain and ion transport as upregulated, and growth-factor/proliferation as downregulated in HeLa:MondoA-KO cells (*Figure 6C*). Together these data show that MondoA is required for the transcriptional response to DMEM$^{Acidic}$ treatment and suggests that MondoA may have an essential role in an adaptive transcriptional response to acidosis.

It is striking that TXNIP and ARRDC4 are the only two genes that show near complete dependence on MondoA in normal medium and the strongest MondoA-dependent induction in response to DMEM$^{Acidic}$ (*Figure 6B*). To investigate this phenomenon further, we used Chromatin Immunoprecipitation Sequencing (ChIP-seq) to determined MondoA's genome-wide occupancy under both media conditions. MondoA occupied ~700 sites in normal pH media and the occupancy of these sites increased following DMEM$^{Acidic}$ treatment. Of these 700 sites, 128 were located within ±2000

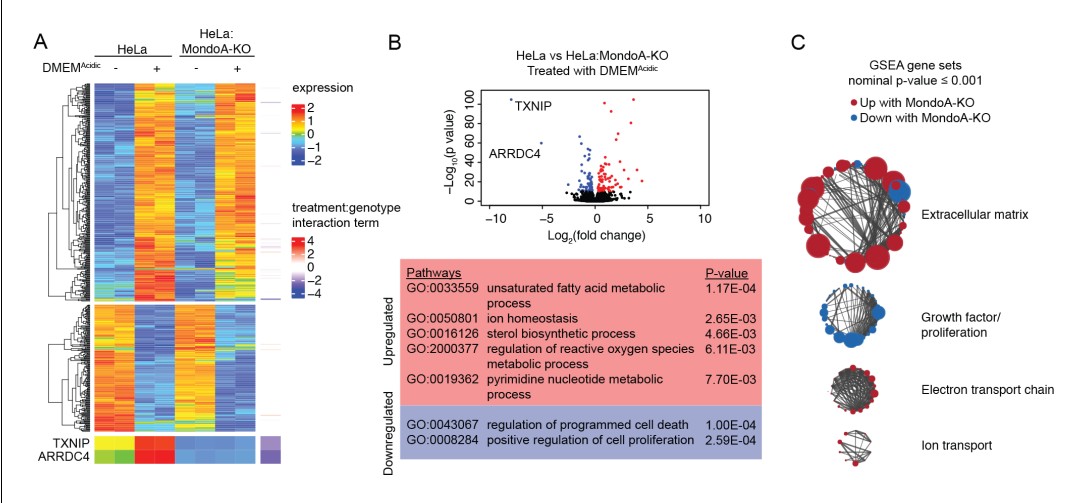

**Figure 6.** The MondoA-dependent acidosis response. RNA-sequencing was used to determine differentially regulated genes for HeLa and HeLa:MondoA-knockout cells treated with DMEM$^{Acidic}$ for 4 hr. Differentially regulated genes from duplicate biological samples were determined. (**A**) Heatmaps depicting TXNIP, ARRDC4 and the top 500 differentially regulated genes in HeLa cells following DMEM$^{Acidic}$. The genotype:treatment interaction term was calculated using DESeq2 and indicates the influence of both genotype and treatment on differential expression. (**B**) Volcano plot of log$_2$(fold-change) of HeLa cells treated with DMEM$^{Acidic}$ compared to HeLa:MondoA-KO cells treated with DMEM$^{Acidic}$. Genes with an adjusted p-value$\leq$1E-10 that are upregulated or downregulated in HeLa:MondoA-KO cells treated with DMEM$^{Acidic}$ are indicated in red and blue, respectively. Overrepresentation analysis was performed for upregulated and downregulated genes. Enriched pathways and their respective p-values are given in the red and blue boxes for upregulated and downregulated genes, respectively. (**C**) GSEA and leading edge analysis was conducted for HeLa cells treated with DMEM$^{Acidic}$ compared to HeLa:MondoA-KO cells treated with DMEM$^{Acidic}$. Depicted are networks of gene sets with a nominal p-value$\leq$0.001. Node colors are representative of whether the gene set was positively (red) or negatively (blue) enriched. Node size represents gene set size. Connecting line thickness represents similarity between two nodes.

DOI: https://doi.org/10.7554/eLife.40199.012

The following source data is available for figure 6:

**Source data 1.** Acidosis regulated genes and pathways.
DOI: https://doi.org/10.7554/eLife.40199.013
**Source data 2.** Pathways enriched in acidosis treated MondoA knockout HeLa cells.
DOI: https://doi.org/10.7554/eLife.40199.014

base pairs of the transcriptional start site of the nearest gene (***Figure 7A***). The binding sites in the promoters of TXNIP and ARRDC4 displayed much higher MondoA occupancy compared to its occupancy of other promoter binding sites (***Figure 7B and C***). We validated our ChIP-sequencing result at four binding sites using ChIP-PCR. Relative to the input controls, the amount of MondoA on the TXNIP promoter was about twice that observed on the ARRDC4 promoter. Relative to the high occupancy of MondoA on the TXNIP promoter, occupancy of MondoA on the KLF10 and TMEM97 promoters was reduced approximately 30 and 60 fold (***Figure 7D***), respectively. Regardless of the amount of MondoA bound at a given promoter in DMEM, the magnitude of increase at all targets following DMEM$^{Acidic}$ treatment was within a fairly narrow range (***Figure 7—figure supplement 1***).

## Discussion

Previous studies established that MondoA's transcriptional activity is highly-dependent on two signals: glucose and a signal from the ETC. By dissecting how low pH drives MondoA transcriptional activity, we establish that the ETC signal is mtATP. Previous studies showed that a functional ETC is required for basal TXNIP expression, thus we propose that mtATP is a general requirement for MondoA transcriptional activity. Via the activity of OMM-bound HK2, mtATP couples to cytoplasmic glucose to generate G6P, which drives the nuclear accumulation and transcriptional activity of MondoA:Mlx complexes (***Figure 8***). Further, we previously demonstrated (***Sans et al., 2006***), and confirmed here (***Figure 4A***), that MondoA and Mlx also interact with the OMM. Therefore, we propose that MondoA:Mlx and HK2 constitute a sensing and response module that integrates signals from the

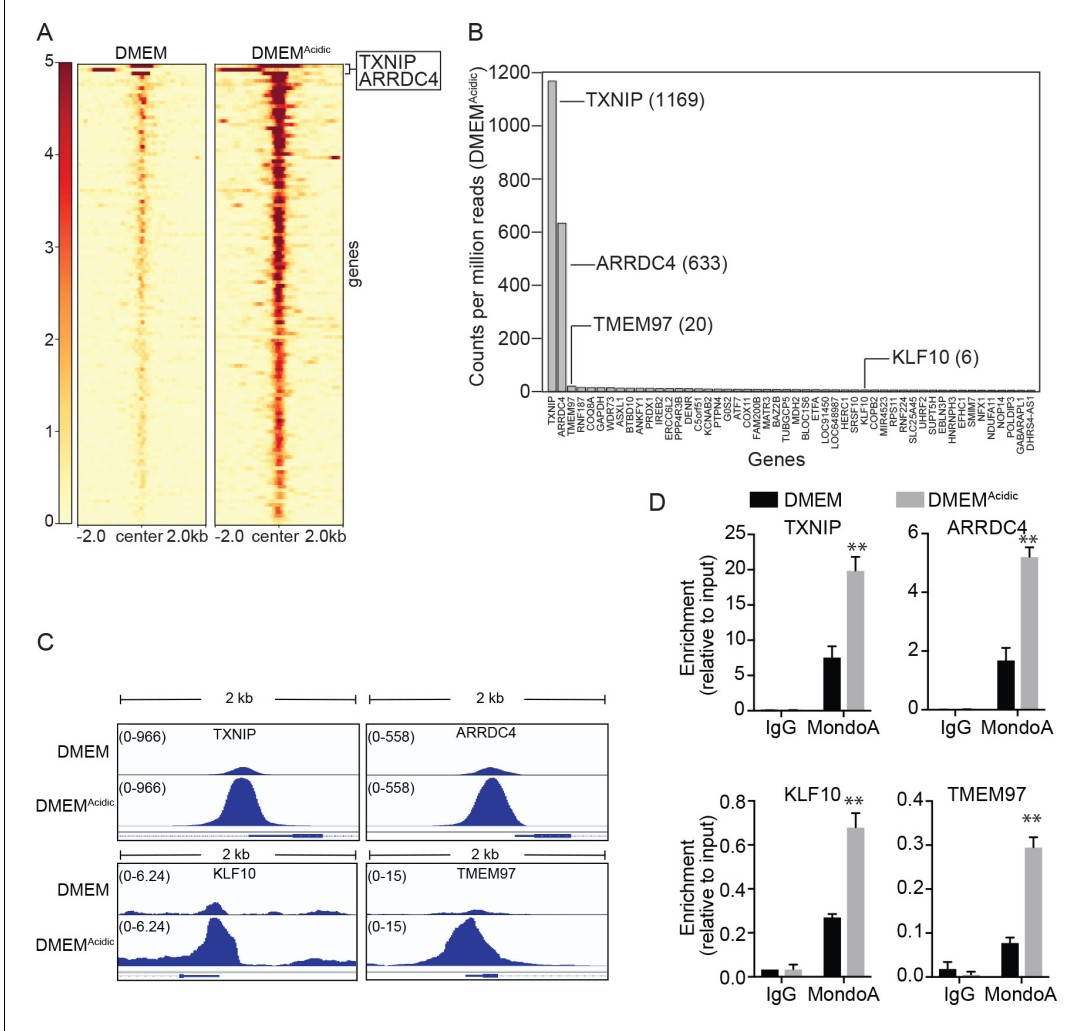

**Figure 7.** MondoA preferentially occupies the promoters of TXNIP and ARRDC4. ChIP sequencing was performed on single biological samples and used to identify MondoA's genomic binding sites in HeLa cells grown in DMEM or treated with DMEM$^{Acidic}$ for 4 hr. (**A**) Heatmaps showing ~100 MondoA binding sites located within the promoters of potentially regulated genes. We defined the promoter region as being 2 kb upstream or downstream of the transcriptional start site (TSS). (**B**) Histogram of counts per million reads (CPM) values for the top 50 promoter-localized MondoA genomic binding sites following a 4 hr DMEM$^{Acidic}$ treatment of HeLa cells. (**C**) Genome browser views derived from ChIP-seq experiment of the MondoA binding sites in the promoters of the indicated genes. (**D**) Independent biological triplicates were grown in DMEM or treated for 4 hr with DMEM$^{Acidic}$. ChIP-PCR was used to validate MondoA occupancy on the promoters of the indicated genes under the two experimental conditions. **p<0.01.

DOI: https://doi.org/10.7554/eLife.40199.015

The following figure supplement is available for figure 7:

**Figure supplement 1.** DMEM$^{Acidic}$ increases MondoA genomic occupancy.

DOI: https://doi.org/10.7554/eLife.40199.016

cytoplasm and the mitochondria to coordinate the transcriptional response to the cell's two predominant energy sources. We studied how acidosis drives mtATP production and MondoA transcriptional activity. It will be interesting to determine whether other cellular signals that drive MondoA transcriptional activity also function by controlling mtATP pools. MondoA also appears to be activated by pH-dependent mechanism that is independent of functional mitochondria (*Figure 2B and G*). There are a number of conserved histidines, which have titratable protons in the physiological range, in MondoA Conserved Region 1 and in the DNA-binding basic region (*McFerrin and Atchley, 2012*), which may account for this effect.

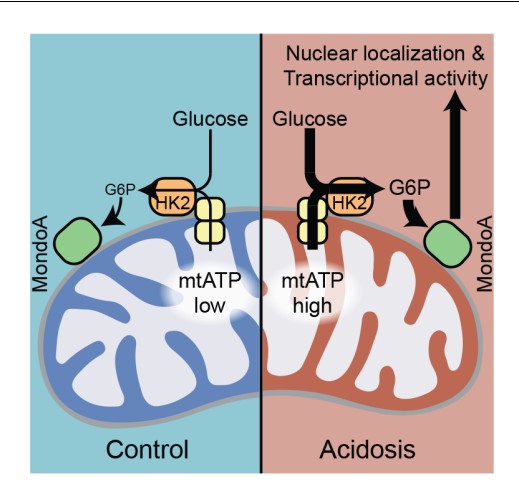

**Figure 8.** MondoA coordinates the transcriptional response to cytoplasmic glucose and mitochondrial ATP. Schematic depicting how acidosis drives MondoA transcriptional activity through the generation of mitochondrial ATP and utilization by mitochondria-bound hexokinase to produce G6P, which drives MondoA nuclear localization and transcriptional activity.

DOI: https://doi.org/10.7554/eLife.40199.017

By binding mitochondria, hexokinase has increased specific activity and decreased feedback inhibition by G6P (*Robey and Hay, 2006*). By localizing to the mitochondria and sensing G6P derived from mitochondria-bound hexokinase, we propose that MondoA activity is coupled to mitochondrial hexokinase activity and mtATP synthesis. Given the OMM localization of MondoA, Mlx and HK2, we suggest that the OMM serves as a scaffold for nutrient sensing by MondoA, akin to other nutrient sensors that are tethered to organellar membranes. For example, the mTORC1 complex is tethered to the lysosome where it integrates intra-lysosomal nutrient levels and cytosolic growth factor signals to control biosynthesis (*Wolfson and Sabatini, 2017*), and the SREBP/Scap complex which is resident in the ER membrane where it monitors cholesterol and oxysterol availability and controls a transcriptional response to low sterol levels (*Moon, 2017*).

Our data suggest that MondoA functions as a coincidence detector which simultaneously senses mtATP and glucose through the synthesis of G6P (*Figure 8*). Such a model ensures that the availability of glucose is tightly linked to mitochondrial activity and ATP synthesis. Conceptually, by coupling mtATP and cytosolic glucose, MondoA functions as a sensor of high cellular energy charge and via its transcriptional regulation of TXNIP, ARRDC4 and potentially other targets, restricts glucose uptake and aerobic glycolysis to restore energy balance. Further, high TXNIP levels are correlated with oxidation of triglycerides, branched chain amino acids and lactate (*DeBalsi et al., 2014*; *Bodnar et al., 2002*), suggesting an additional role for the MondoA/TXNIP axis in driving utilization of non-glucose fuels when cellular energy charge is high. The precise mechanistic details of how MondoA senses G6P and the impact on cell metabolism remains to be clarified; however, the current data is most consistent with a allosteric model where G6P binds MondoA directly (*McFerrin and Atchley, 2012*; *Peterson et al., 2010*; *Li et al., 2010*).

Changes in intracellular pH have dramatic effects on cell function: generally acidic pH is anti-proliferative whereas alkaline pH is pro-proliferative. Acidic intercellular pH is correlated with an inhibition of aerobic glycolysis and a blockage of glucose uptake (*Webb et al., 2011*). Our data suggest that TXNIP induction contributes to this suppression of glucose metabolism driven by acidic pH. Further, an acidosis-dependent gene signature, of which TXNIP is a member correlates with better clinical outcomes in breast cancer (*Chen et al., 2010*; *Chen et al., 2008*). Interestingly, Otto Warburg noted that increased sodium bicarbonate and alkaline pH favor glycolysis (*Koppenol et al., 2011*; *Warburg, 1925*). Intracellular alkalization is now a widely accepted hallmark of cancer metabolism (*Webb et al., 2011*), that has pleiotropic effects on tumorigenesis, the most predominant being a transition from oxidative metabolism to aerobic glycolysis (*Reshkin et al., 2000*). Because, MondoA transcriptional activity at the TXNIP promoter is suppressed by high sodium bicarbonate and alkaline pH (*Figure 1—figure supplement 2B and C*), we propose that TXNIP down regulation contributes to the shift to high glucose uptake driven by alkaline pH. It will be important to determine whether alkaline pH suppresses TXNIP expression by restricting mtATP production. Collectively, our data suggest that the MondoA/TXNIP axis plays a critical role in how cancer cells sense and respond to dysregulated pH.

Our gene expression data demonstrates that MondoA is essential for the regulation of 37% of an acidosis-driven transcriptional response. Among the MondoA-dependent genes are fatty acid and mitochondrial metabolism genes. Given that these pathways are enhanced by acidosis

(*Corbet et al., 2016*; *Lamonte et al., 2013*; *Khacho et al., 2014*), we propose that MondoA plays a critical role in an adaptive metabolic response to acidosis. Our ChIP-seq and ChIP-PCR analysis revealed the MondoA binds the TXNIP and ARRDC4 promoters at least an order of magnitude more strongly than other promoter binding sites. This likely accounts their high MondoA-dependence at baseline and following DMEM$^{Acidic}$ treatment. An enrichment of high affinity E-boxes in the promoters of the TXNIP and ARRDC4 promoters is apparently not sufficient to account for their high occupancy by MondoA (data not shown). The relative increase in MondoA occupancy following DMEM$^{Acidic}$ treatment is similar at all of its binding sites, further supporting the idea that features outside the primary sequence of the binding site, such as chromatin modifications or ancillary chromatin-binding factors must contribute to high affinity MondoA binding. Together these data suggest that TXNIP and ARRDC4 are the primary MondoA effectors in the cellular transcriptional response to low pH.

Finally, it is well established that oncogenes drive a shift from oxidative metabolism to aerobic glycolysis (*Pavlova and Thompson, 2016*). The resulting shift away from ATP synthesis in the mitochondria to ATP synthesized by glycolysis in the cytosol would be predicted to restrict MondoA-dependent activation of TXNIP expression and reinforce glucose uptake and aerobic glycolysis. Strikingly, TXNIP expression is also downregulated by a variety of pro-growth signals such as mTOR, PI3K, Ras and Myc (*Elgort et al., 2010*; *Kaadige et al., 2015*; *Shen et al., 2015*), which results in increased glucose uptake. Together these two findings place MondoA and its regulation of TXNIP both upstream and downstream of metabolic reprogramming towards aerobic glycolysis.

## Materials and methods

**Key resources table**

| Reagent type (species) or resource | Designation | Source or reference | Identifiers | Additional information |
|---|---|---|---|---|
| Antibody | Anti-HK2 (goat polyclonal) | Santa Cruz | sc6521 | (1:1,000) |
| Antibody | Anti-MLX (rabbit monoclonal (D8G6W) | Cell Signaling | 85570S | (1:1,000) |
| Antibody | Anti-MlxIP (MondoA) (rabbit polyclonal) | Proteintech | 13614–1-AP | (1:2,000) |
| Antibody | Anti-SDHA (2E3GC12FB2AE2) (mouse monoclonal | Abcam | AB147 | (1:15,000) |
| Antibody | Anti-Tubulin (mouse monoclonal) | Molecular Probes | 236–10501 | (1:50,000) |
| Antibody | Anti-TXNIP (rabbit monoclonal) | Abcam | ab188865 | (1:2,000) |
| Antibody | Anti-LaminB1 (rabbit polyclonal) | Abcam | ab16048 | (1:1,000) |
| Antibody | anti-goat IgG-HRP (donkey polyclonal) | Santa Cruz | sc-2056 | (1:2,000) |
| Antibody | Mouse IgG, HRP-linked whole Ab (sheep polyclonal) | GE Life Science | NA-931 | (1:5,000) |
| Antibody | Rabbit IgG, HRP-linked whole Ab (donkey polyclonal) | GE Life Science | NA-934 | (1:15,000) |
| Chemical compound, drug | BCECF-AM | Thermo Fisher | B1170 | |
| Chemical compound, drug | Blotting Grade Blocker Non-fat Dry Milk | Bio-Rad | 1706404XTU | |
| Chemical compound, drug | CCCP | Sigma Aldrich | C2759 | |

*Continued on next page*

*Continued*

| Reagent type (species) or resource | Designation | Source or reference | Identifiers | Additional information |
|---|---|---|---|---|
| Chemical compound, drug | Chloroquine | Sigma Aldrich | 415480 | |
| Chemical compound, drug | Deoxy-D-Glucose, 2-[1,2-3H(N)] | American Radiolabeled Chemicals, Inc. | 0103–250 | |
| Chemical compound, drug | DMEM | Gibco | 11995–065 | |
| Chemical compound, drug | DMEM Powder without sodium bicarbonate, glucose, L-glutamine, sodium pyruvate and phenol red | Cellgro | 90–113-PB | |
| Chemical compound, drug | DMSO | Fisher | BP231 | |
| Chemical compound, drug | FCCP | Sigma Aldrich | C2920 | |
| Chemical compound, drug | Fetal bovine serum (FBS) | Gibco | 26140–079 | |
| Chemical compound, drug | Glucose | Fisher | D16-1 | |
| Chemical compound, drug | Glutamine | Cellgro | 25–005 Cl | |
| Chemical compound, drug | HBSS | Gibco | 24020–117 | |
| Chemical compound, drug | HEPES | Sigma Aldrich | H3375 | |
| Chemical compound, drug | JC1 | Thermo Fisher | T3168 | |
| Chemical compound, drug | Metformin | Sigma Aldrich | D150959 | |
| Chemical compound, drug | Monensin | Sigma Aldrich | M5273 | |
| Chemical compound, drug | Non-essential amino acids | Gibco | 11140–050 | |
| Chemical compound, drug | Oligomycin A | Sigma Aldrich | 75351 | |
| Chemical compound, drug | Pennicillin/Streptomycin | Gibco | 15140–112 | |
| Chemical compound, drug | Phenol Red | Sigma Aldrich | P-0290 | |
| Chemical compound, drug | Sodium bicarbonate | Fisher | L-23200 | |
| Chemical compound, drug | Sodium pyruvate | Gibco | 11360–070 | |
| Chemical compound, drug | Trypsin-EDTA (0.25%) | Gibco | 25200–056 | |
| Chemical compound, drug | Tween-20 | Fisher | BP-337 | |
| Commercial assay, or kit | Quick RNA miniprep kit | Genesee Scientific | R1055 | |
| Commercial assay, or kit | ATP determination kit | Thermo Fisher | A22066 | |
| Commercial assay, or kit | Mitochondria isolation kit for cultured cells | Thermo Fisher | 89874 | |

*Continued on next page*

*Continued*

| Reagent type (species) or resource | Designation | Source or reference | Identifiers | Additional information |
|---|---|---|---|---|
| Commercial assay, or kit | Stranded mRNA-Seq kit with mRNA capture beads | Kapa Biosystems | KK8421 | |
| Commercial assay, or kit | Galacto-Light Reaction Buffer Diluent with Galacton-Plus | Thermo Fisher | T1055 | |
| Commercial assay, or kit | Luciferase Assay System | Promega | E4550 | |
| Commercial assay, or kit | ProSignal Pico ECL | Genesee Scientific | 20-300B | |
| Commercial assay, or kit | Reporter 5X Lysis Buffer | Promega | E4030 | |
| Commercial assay, or kit | SuperSignal West Femto | Thermo Fisher | 34094 | |
| Cell line (*M. musculus*) | MondoA $^{+/+}$ mouse embryonic fibroblasts | *Peterson et al. (2010)*, PMID: 20385767 | | |
| Cell line (*M. musculus*) | MondoA $^{\Delta/\Delta}$ mouse embryonic fibroblasts | *Peterson et al. (2010)*, PMID: 20385767 | | |
| Cell line (*H. sapiens*) | 143B | Weinberg et al. 2010, PMID: 20421486 | | |
| Cell line (*H. sapiens*) | 143B$\rho^0$ | Weinberg et al. 2010, PMID: 20421486 | | |
| Cell line (*H. sapiens*) | 143B$\rho^0$:Wild type cybrid | Weinberg et al. 2010, PMID: 20421486 | | |
| Cell line (*H. sapiens*) | 143B$\rho^0$:ΔATP6/ΔATP8 cybrid | *Boominathan et al. (2016)*, PMID: 27596602 | | |
| Cell line (*H. sapiens*) | 143B$\rho^0$:ΔATP6/ΔATP8 cybrid + ATP6$_{nuc}$+ATP8$_{nuc}$ | *Boominathan et al. (2016)*, PMID: 27596602 | | |
| Cell line (*H. sapiens*) | HeLa | ATCC | CCL-2 | |
| Cell line (*H. sapiens*) | BJ-Tert | ATCC | CRL-4001 | |
| Sequence-based reagent | TXNIP_forward (human): TGACTTTGGCCTACAGTGGG | *Peterson et al. (2010)*, PMID: 20385767 | | |
| Sequence-based reagent | TXNIP_reverse (human): TTGCGCTTCTCCAGATACTGC | *Peterson et al. (2010)*, PMID: 20385767 | | |
| Sequence-based reagent | TXNIP_forward (mouse): CCTGACCTAATGGCACC | *Peterson et al. (2010)*, PMID: 20385767 | | |
| Sequence-based reagent | TXNIP_reverse (mouse): GAGATGTCATCACCTTCAC | *Peterson et al. (2010)*, PMID: 20385767 | | |
| Sequence-based reagent | ATP5I_forward: CAGGTCTCTCCGCTCATCAAG | This paper | | |
| Sequence-based reagent | ATP5I_reverse: GCCCGAGGTTTTAGGTAATTGT | This paper | | |
| Sequence-based reagent | Actin_forward: TCCATCATGAAGTGTGACGT | *Peterson et al. (2010)*, PMID: 20385767 | | |
| Sequence-based reagent | Actin_reverse: TACTCCTGCTTGCTGATCCAC | *Peterson et al. (2010)*, PMID: 20385767 | | |
| Sequence-based reagent | TXNIP_forward ChIP primer: CAGCGATCTCACTGATTG | This paper | | |
| Sequence-based reagent | TXNIP_reverse ChIP primer: AGTTTCAAGCAGGAGGCG | This paper | | |

*Continued on next page*

*Continued*

| Reagent type (species) or resource | Designation | Source or reference | Identifiers | Additional information |
|---|---|---|---|---|
| Sequence-based reagent | ARRDC4_forward ChIP primer: TGCTTTAGCGAGAACCCAGT | This paper | | |
| Sequence-based reagent | ARRDC4_reverse ChIP primer: TGGACAGACAGTGGGAAACA | This paper | | |
| Sequence-based reagent | TMEM97_forward ChIP primer: CTTACTGCAGAAGGCCCAAG | This paper | | |
| Sequence-based reagent | TMEM97_reverse ChIP primer: TGTAGATTGCGGTTGTGAGC | This paper | | |
| Sequence-based reagent | KLF10_forward ChIP primer: AATCAACGGCAAAGGTGTGT | This paper | | |
| Sequence-based reagent | KLF10_reverse ChIP primer: CACTCAATCAGGTGGCCTCT | This paper | | |
| Sequence-based reagent | siRNA: Dharmacon ON-TARGETplus control siRNA | GE Life Sciences | D00-1810-10-20 | |
| Sequence-based reagent | siRNA: siATP5I SmartPool | GE Life Sciences | M-019688–01 | |
| Sequence-based reagent | siRNA: siSLC25A5 SmartPool (siANT2) | GE Life Sciences | M-007486 | |
| Sequence-based reagent | siRNA: siHK2 SmartPool | GE Life Sciences | L-006735-00-0005 | |
| recombinant DNA reagent | LXSH (plasmid) | *Stoltzman et al. (2008)*, PMID: 18458340 | | |
| Recombinant DNA reagent | LXSH-MondoA (plasmid) | *Stoltzman et al. (2008)*, PMID: 18458340 | | |
| Recombinant DNA reagent | LXSH-MondoA(I766P) (plasmid) | *Stoltzman et al. (2008)*, PMID: 18458340 | | |
| Recombinant DNA reagent | pcDNA3-AT1.03 (ATEAM) (plasmid) | *Imamura et al. (2009)*, PMCID: PMC2735558 | | |
| Recombinant DNA reagent | pcDNA3-mitAT1.03 (Mit-ATEAM) (plasmid) | *Imamura et al. (2009)*, PMID: 19720993 | | |
| Recombinant DNA reagent | pcDNA3-AT1.03 R122K/R126K (plasmid) | *Imamura et al. (2009)*, PMID: 19720993 | | |
| Recombinant DNA reagent | pcDNA3-mitAT1.03 R122K/R126K (plasmid) | *Imamura et al. (2009)*, PMID: 19720993 | | |
| Recombinant DNA reagent | pEGFP-N1-mVDAC1 (plasmid) | *Zaid et al. (2005)*, PMID: 15818409 | | |
| Recombinant DNA reagent | pEGFP-N1-mVDAC1 (E72Q) (plasmid) | *Zaid et al. (2005)*, PMID: 15818409 | | |
| Recombinant DNA reagent | pCDV-SPORT6-HK2 (plasmid) | *Stoltzman et al. (2008)*, PMID: 18458340 | | |
| Recombinant DNA reagent | pCDV-SPORT6-HK2 (D657A) (plasmid) | *Stoltzman et al. (2008)*, PMID: 18458340 | | |
| Recombinant DNA reagent | pcDNA3.1-mVDAC1-GFP(1-10) (plasmid) | This paper | | Progenitors: PCR, pcDNAGFP(1-10) |
| Recombinant DNA reagent | pcDNA3.1-mVDAC1 (E72Q)-GFP(1-10) (plasmid) | This paper | | Progenitors: PCR, pcDNAGFP(1-10) |

*Continued on next page*

*Continued*

| Reagent type (species) or resource | Designation | Source or reference | Identifiers | Additional information |
|---|---|---|---|---|
| Recombinant DNA reagent | pcDNA3.1-HK2-GFP(11) (plasmid) | This paper | | Progenitors: PCR, pcDNAGFP(11) |
| Recombinant DNA reagent | pGL3Basic-TXNIP_Promoter (plasmid) | *Peterson et al. (2010)*, PMID: 20385767 | | |
| Recombinant DNA reagent | pGL3Basic-TXNIP_Promoter(ChoRE$_{mut}$) (plasmid) | *Peterson et al. (2010)*, PMID: 20385767 | | |
| Recombinant DNA reagent | pcDNAGFP(1-10) | *Kamiyama et al. (2016)*, PMID: 26988139 | | |
| Recombinant DNA reagent | pcDNAGFP(11) | *Kamiyama et al. (2016)*, PMID: 26988139 | | |
| Software, algorithm | Prism | Graphpad Software | | |
| Software, algorithm | ImageJ | https://imagej.nih.gov/ij/ | | |
| Software, algorithm | CFX Manager 3.1 | Bio-Rad | | |
| Software, algorithm | R | https://www.r-project.org | | |
| Software, algorithm | javaGSEA | Broad Institute | | |
| Software, algorithm | Cytoscape 3.6.1 | https://cytoscape.org | | |
| Software, algorithm | NIS Elements | Nikon | | |
| Other | Nunc Lab-Tek II Chambered Coverglass, 8-well | Thermo Fisher | 155409PK | |
| Other | 3.5 mm glass bottom culture dishes | MatTek Corporation | P35G-.15–14 C | |
| Other | Hybond P PVDF Membrane; 0.45 µm | Genesee Scientific | 83–646R | |
| Other | 2 ml PTFE tissue grinder | VWR | 89026–398 | |
| Other | Bioruptor Plus sonication devise | Diagenode | B01020001 | |

## Cell lines

A list of cell lines used is provided in the Key Resources Table. All cells were maintained in DMEM + 10% FBS (Gibco), 100 units/mL penicillin (Gibco) and 100 units/mL streptomycin (Gibco). 143Bρ⁰ and cybrids were cultured with 1 mM sodium pyruvate and 50 µg/mL uridine. Cells were passaged and treated in an incubator set at 37°C and 5% $CO_2$.

HeLa:MondoA-KO cells were generated by expressing CRISPR/Cas9, three sgRNAs (GeCKO library 2.0) and a homology-directed repair (HDR) construct containing a puromycin-resistance cassette (Santa Cruz Biotechnology). HDR incorporation into the genome was determined by selecting for cells resistant to 2.5 µg/mL puromycin. Loss of MondoA was determined by immunoblotting.

## Treatments

HBSS was supplemented with glucose to 20 mM prior to treatment. Low pH treatment media (pH 6.5) was prepared from DMEM powder without glutamine, glucose, pyruvate, sodium bicarbonate and phenol red (DMEM$^{Acidic}$). The following were added: glutamine to 2 mM, glucose to 20 mM, pyruvate 1 mM, sodium bicarbonate to 0.35 g/L and phenol red to 16 mg/L. For live cell imaging, phenol red was omitted.

## Plasmid construction

Plasmids were created using either standard restriction digest and ligation or Gibson assembly (NEB). A list of plasmids used, the vector backbone and their source is provided in the Key Resources Table.

## Quantitative PCR

Total cellular RNA was extracted using a Quick RNA Miniprep Kit (Zymo Research) according to manufacturer's recommendations. cDNA was synthesized from 200 ng mRNA using the GoScript Reverse Transcription System (Promega) with oligo-dT primers. A 100-fold dilution was used in a PCR reaction containing SYBR Green and analyzed on a CFX Connect Real Time System. Values were determined using a standard curve. For each sample, three technical replicates were performed and averages determined.

## Immunoblotting

Equal concentrations of denatured protein lysates were resolved on 10% SDS-PAGE gel with a stacking gel. Proteins were electrotransferred to PDVF membrane (Genesee Scientific). Membranes were incubated in 5% (weight/volume) blotting-grade non-fat dry milk (Bio-Rad) in TBST (Tris-buffered saline, pH 7.4% and 0.1% Tween-20) for 30 min at room temperature with gentle rocking. Membranes were then transferred to antibody-dilution buffer (20 mM Tris, pH 8.0; 200 mM NaCl; 0.25% Tween-20; 2% bovine serum albumin; 0.1% sodium azide) and incubated for one hour at room temperature or overnight at 4°C with gentle rocking. Membranes were washed with TBST and vigorous rocking at room temperature. Membranes were then incubated in secondary antibody diluted in 5% (weight/volume) blotting-grade non-fat dry milk (Bio-Rad) in TBST for one hour at room temperature with gentle rocking. Membranes were then washed again and proteins were detected with chemiluminescence using standard or high sensitivity ECL (Genesee Scientific or Thermo Fisher, respectively).

## ATP quantification

After treatment, cells were washed once with cold PBS. Cells were scraped into boiling TE buffer (1 mL per 3.5 cm dish), which was collected into 1.5 mL centrifuge tube. Cells were then boiled for 5 min. Lysates were cleared by centrifugation at 20,000xg for 5 min. An ATP determination kit (Thermo Fisher) was used with 10 μL of supernatant. A standard curve was generated using purified ATP.

## Live cell imaging (mtATP determination): Widefield microscopy

Widefield microscopy was used for *Figure 3* and *Figure 3—figure supplement 2*. Cells were plated on 3.5 mm glass bottom culture dishes (MatTek Corporation). The following day 100 ng Mit-ATEAM was transfected using Lipofectamine 3000 (Thermo Fisher) according to manufacturer's recommendations. The next day cells were treated with DMEM$^{Acidic}$ lacking phenol red. Real time live imaging was conducted for 8 hr using a Nikon A1R with a 40X lens. For each time point images were captured using 488/525 (YFP), 405/480 (CFP), and 405/525 (FRET) excitement/emission (nm).

Images were analyzed using ImageJ. We used the YFP channel to identify and isolate mitochondrial regions for each image. We isolated these same regions from the CFP and FRET channel. Total intensity was determined for each image. FRET/CFP ratios were determined and normalized to the 2 hr time point. RatioPlus was used to make pseudo-colored images.

## Live cell imaging (mtATP determination): Confocal microscopy

Confocal microscopy was used for *Figure 3—figure supplement 2A and C*. Cells were plated on 8-well Nunc Lab-Tek II Chambered Coverglass (Thermo Fisher). The following day 200 ng DNA was transfected using Lipofectamine 3000 (Thermo Fisher) according to manufacturer's recommendations. The next day cells were treated DMEM$^{Acidic}$ lacking phenol red or DMEM lacking phenol red with CCCP. Real time live imaging was conducted for 8 hr using a Nikon A1 with a 20X lens. For each time point images were captured using 402/488 (CFP) and 402/525 (FRET) excitement/emission (nm). Images were analyzed using ImageJ. RatioPlus was used to make pseudo-colored images. Total intensity for each image was determined.

## Glucose uptake

Cells were incubated with deoxy-D-glucose-2[1,2-3H(N)] (American Radiolabeled Chemicals, Inc) in Krebs-Ringer-HEPES buffer (NaCl,116 mM; KCl, 4 mM; MgCl$_2$, 1 mM; CaCl$_2$, 1.8 mM; 2-deoxy-D-glucose, 20 mM; HEPES pH 7.4, 10 mM) for 10 min. Cells were then washed, harvested and analyzed

for radioactivity using a scintillation counter. A standard was generated to determine the exact molar content in each sample. Deoxy-D-glucose-2[1,2-3H(N)] was normalized to protein content as determined by a Bradford Protein Assay (Bio-Rad).

## Mitochondrial membrane potential

Cells were plated on 8-well Nunc Lab-Tek II Chambered Coverglass (Thermo Fisher). 45 min prior to the experiment, cells were loaded with JC1 (1 µg/mL). Cells were treated DMEM$^{Acidic}$ lacking phenol red or DMEM lacking phenol red. A Nikon A1 confocal and NIS Elements AR were used to capture images. For each time point images were captured by exciting with 488 nm light and reading the emission at 530 nm (green) and 595 nm (red). The ratio of red to green was used to quantify changes in membrane potential one hour after DMEM$^{Acidic}$ treatment.

## Intracellular pH

Cells were plated on 3.5 mm glass bottom culture dishes (MatTek Corporation). The next day cells were treated with normal or low pH DMEM for 4 hr. Cells were treated with BCECF-AM (1 µM) 30 min prior to the end of the experiment. A standard curve was generated by treating cells with media of varying pHs and Nigericin (5 µM), which equilibrates intracellular and extracellular pH. A Nikon A1 confocal and NIS Elements AR were used to capture images by exciting with 488 nm and reading emission at 530 nm and 595 nm. The 595/530 nm fluorescence emission ratio was used to generate a calibration curve and determine intracellular pH for acidosis-treated cells.

## Mitochondria purification

Mitochondria were purified from ~20 × 10$^6$ cells using a Mitochondria Isolation Kit for Cultured Cells (Thermo Fisher). Cells were processed using a PTFE tissue grinder (VWR). Following purification, mitochondria were resuspended in 100 µl radioimmunoprecipitation (RIPA) buffer. 100 µl of both mitochondria and cytosolic fractions were sonicated at using a Bioruptor sonication device (Diagenode). Sonication was performed 4°C using 30 s on/off pulses at the high setting. Following sonication, lysates were centrifuged and supernatants were collected and analyzed for protein content using a Bradford Protein Assay (Bio-Rad). 1–5 µg of sample were used for immunoblot analysis.

## Subcellular fractionation: Nuclei and cytoplasm

Three days prior to fractionation siRNA pools specific ANT2, ATP5I and HKII were transfected using Lipofectamine 3000 (Thermo Fischer). Cells were washed with cold PBS and dislodged from plate by scraping. Cells were pelleted by centrifugation and resuspended in 1 mL of fractionation buffer (40 mM HEPES pH 7.9, 137 mM NaCl, 2.7 mM KCl, 1.5 mM MgCl$_2$, 0.34 M sucrose, 10% glycerol, 1 mM DTT, 0.5% NP40, protease and phosphatase inhibitors). Cells were incubated on ice for 10 min then pelleted by centrifugation at 1000 rcf for 5 min. The supernatant was kept (cytoplasm) and the pellet (nuclei) was washed three times with 0.5 mL fractionation buffer.

## Luciferase assay

Cells were seeded and the next day transfected with constructs containing a 1518 bp fragment of the TXNIP promoter (or a mutant)-driven luciferase and CMV-driven beta-galactosidase (*Kaadige et al., 2009*). Cells were harvested in 1X Buffer RLB (Promega). Luciferase was detected using the Luciferase Detection System (Promega) and beta-galactosidase was detected using Galacto-Light Reaction Buffer Diluent with Galacto-Plus Substrate (Thermo Fisher). Luminescence was determined using a GloMax 96 Microplate Luminometer (Promega). Luciferase values were normalized to beta-galactosidase values.

## GC-MS

Following treatment, cells were collected into a 1.5 mL microcentrifuge tube then snap frozen using liquid nitrogen. Cells were kept at −80°C until metabolite extraction was performed. 450 µL of cold 90% methanol and internal standards were added to cells and incubated at −20°C for 1 hr. Tubes were then centrifuged at −20,000 × g for 5 min at 4°C. Supernatants were dried using a speed-vac.

Samples were converted into volatile derivatives amenable to GC-MS. Briefly, dried samples were resuspended in O-methoxylamine hydrochloride (40 mg/mL) then mixed with 40 µL N-methyl-N-

trimethylsilyltrifluoracetamide and mixed at 37°C. After incubation, 3 μL fatty acid methyl ester standard solution was added. 1 μL of this final solution was injected into gas chromatograph with an inlet temperature of 250°C. A 10:1 split ratio was used. Three temperatures were ramped with a final temperature of 350°C and a final 3 min incubation. A 30 m Phenomex ZB5-5 MSi column was used. Helium was used as carrier gas at 1 mL/minute. Samples were analyzed again with a 10-fold dilution.

Data was collected using MassLynx 4.1 software (Waters). Metabolites were identified and peak area was determined using QuanLynx. Data was normalized using Metaboanalyst 3.6 (http://www.metaboanalyst.ca/). Quantile normalization, log transformation and Pareto scaling were used. Normal distribution of values was used to determine fold changes.

## RNA-sequencing library construction and analysis

Total RNA was extracted from cells using a Quick RNA Miniprep Kit (Zymo Research) according to manufacturer's recommendations. mRNA was isolated and library production performed using a Stranded mRNA-Seq Kit with mRNA Capture Beads (Kapa). Library quality was analyzed using an Agilent High Sensitivity D1000 ScreenTape. Single-end sequencing for 50 cycles was performed using an Illumina HiSeq. The resulting FASTQ files were aligned to the human genome (hg38) using STAR. DESeq2 was used to quantify transcript abundance, differential expression, FPKM values, and interaction terms (genotype:treatment combinatorial statistic).

Overrepresentation analysis was performed using ConsensusPathDB. Pathway-based sets were analyzed from Wikipathways. A p-value cutoff of 0.01 and a minimum overlap of 2 genes was used. Enriched pathways were verified by comparing fold-changes obtained from DESeq2.

Gene set enrichment analysis and leading-edge analysis (Broad Institute) was conducted using FPKM values and all gene sets from in the Molecular Signature Database. Leading-edge analysis was visualized using the Cytoscape (p-value≤0.001 and overlap coefficient ≥0.5).

## Chromatin Immunoprecipitation-sequencing (ChIP-seq)

HeLa cells were grown in DMEM with neutral pH and acidic DMEM for 4 hr. After 4 hr, 20 million HeLa cells were crosslinked with 1% formaldehyde for 10 min at room temperature and then were treated with 0.125M glycine for 5 min to quench cross-linking reaction. Crosslinked cells were washed with 1X cold PBS. After that, cells were lysed and scraped with Farnham lysis buffer (5 mM PIPES pH 8.0, 85 mM KCl, 0.5% NP-40) to harvest. Cells were resuspended in RIPA buffer (1X PBS, 1% NP-40, 0.5% sodium deoxycholate, 0.1% SDS) for sonication. Sonication were performed on an Active Motif EpiShear Probe Sonicator for 5 min cycles of 30 s on and 30 s off with 40% amplitude. For immunoprecipitation, 5 μg MondoA (Proteintech) antibody was used. Immunocomplexes were captured with Dynabeads M-280 sheep anti-rabbit (Invitrogen). Input for each condition was served as control. ChIP was performed as previously described (*Reddy et al., 2009*). Briefly, blunted ChIP DNA fragments were ligated with sequencing adapters. The ligated ChIP DNA fragments were amplified with library PCR primers that contain barcodes for 15 cycles. ChIP libraries were sequenced using Illumina HiSeq Sequencing with 50 cycles of single-read. The resulting Fastq files were aligned to the human genome (hg19) using NovoAlign. Peaks were called using Model-Based Analysis of ChIP-seq-2 (MACS2) (*Zhang et al., 2008*) using a p-value cut-off of 0.01 and the mfold parameter between 5 and 50. Heatmap was generated using deepTools 2.0. MondoA-bound genes were annotated using ChIPseeker, an R package (*Yu et al., 2015*).

## Chromatin immunoprecipitation coupled with quantitative PCR (ChIP-qPCR)

HeLa cells were grown in DMEM with neutral pH and acidic DMEM for 4 hr. After 4 hr, crosslinked and sheared chromatin was prepared as described in ChIP-seq. Input and IgG for each condition were served as the control. For Immunoprecipitation, 5 μg MondoA (Proteintech, 13614–1-AP) and IgG (invitrogen, 026162) antibodies were used. Immunocomplexes were captured with Dynabeads M-280 sheep anti-rabbit (Invitrogen, 11203D). After reversal of crosslinks and ChIP DNA purification (Zymo Research, D5201), MondoA binding on TXNIP, ARRDC4, KLF10 and TMEM97 were determined by normalizing to input. Primers used for qPCR:

## Gene signature

mRNA expression z-scores were obtained for 2509 breast cancer tumors (*Pereira et al., 2016*). Acidosis regulated genes were determined from the gene set GO_RESPONSE_TO_ACIDIC_PH in the Molecular Signature Database. Principal component analysis was conducted for all tumors using the expression levels of acidosis regulated genes. Gene signature scores were determined as the first principle component. This was compared to TXNIP expression for the same tumors.

The normalized expression ($\log_2$(normalized-counts +1)) of TXNIP, SLC16A3 (MCT4), SLC16A1 (MCT1) and SLC9A1 (NHE1) was determined using the UCSC Xena browser. Spearman and Pearson coefficients were used to correlate gene expression. The following datasets were used: TCGA-BRCA, TCGA-LUNG, TCGA-GBM, GTEx-muscle and GTEx-skin.

## Quantification and statistical analysis

Data is presented as mean ±standard deviation. One-way ANOVA was used to account for variation and significance was determined using a two-tailed Student's t-test. Unless otherwise indicated, at least three biological replicates were used for each analysis.

# Acknowledgements

We thank members of the Ayer Lab, Mahesh B Chandrasekharan and Michael Engel for helpful discussions and comments on this manuscript. We also thank Hiroyuki Noji (University of Tokyo), Varda Shoshan-Barmatz (Ben-Gurion University of Negev), Matthew S O'Connor (SENS Research Foundation Research Center), and Jared Rutter (University of Utah) for reagents and advice. We thank Jeff Vahrenkamp and Jay Gertz with help with the bioinformatics approaches used to analyze the ChIP-seq data. DEA was supported by National Institutes of Health Grants 5RO1GM055668 and 1R01CA222650-01, by developmental funds from the Huntsman Cancer Foundation, and by Department of Defense Grant W81XWH1410445.

# Additional information

### Funding

| Funder | Grant reference number | Author |
| --- | --- | --- |
| National Institutes of Health | 5R01GM055668-18 | Donald E Ayer |
| National Institutes of Health | 1R01CA222650-01 | Donald E Ayer |
| U.S. Department of Defense | W81XWH1410445 | Donald E Ayer |
| Huntsman Cancer Foundation | | Donald E Ayer |

The funders had no role in study design, data collection and interpretation, or the decision to submit the work for publication.

### Author contributions

Blake R Wilde, Conceptualization, Writing—original draft, Writing—review and editing, Investigation; Zhizhou Ye, Conceptualization, Data curation, Formal analysis, Investigation, Methodology, Writing—original draft, Writing—review and editing; Tian-Yeh Lim, Conceptualization, Investigation; Donald E Ayer, Conceptualization, Formal analysis, Investigation, Writing—review and editing, Funding aquisition, Project adminsitration

### Author ORCIDs

Blake R Wilde (iD) https://orcid.org/0000-0002-8372-3954
Donald E Ayer (iD) http://orcid.org/0000-0002-5595-3269

### Decision letter and Author response

Decision letter https://doi.org/10.7554/eLife.40199.024
Author response https://doi.org/10.7554/eLife.40199.025

## Additional files

### Supplementary files

• Transparent reporting form
DOI: https://doi.org/10.7554/eLife.40199.018

### Data availability

RNA Sequencing and ChIP sequencing data has been deposited to GEO under GSE117622 and GSE125089.

The following datasets were generated:

| Author(s) | Year | Dataset title | Dataset URL | Database and Identifier |
|---|---|---|---|---|
| Wilde BR, Ye Z, Lim TY, Ayer DE | 2019 | Cellular acidosis triggers human MondoA transcriptional activity by driving mitochondrial ATP production | https://www.ncbi.nlm.nih.gov/geo/query/acc.cgi?acc=GSE117622 | NCBI Gene Expression Omnibus, GSE117622 |
| Wilde BR, Ye Z, Lim TY | 2019 | Cellular acidosis triggers human MondoA transcriptional activity by driving mitochondrial ATP production | https://www.ncbi.nlm.nih.gov/geo/query/acc.cgi?acc=GSE125089 | NCBI Gene Expression Omnibus, GSE125089 |

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
