## [Decision Letter]

Thank you for submitting your article "Cellular acidosis triggers MondoA transcriptional activity by driving mitochondrial ATP production" for consideration by *eLife*. Your article has been reviewed by two peer reviewers, one of whom is a member of our Board of Reviewing Editors, and the evaluation has been overseen by Jonathan Cooper as the Senior Editor. The following individual involved in review of your submission has agreed to reveal his identity: Chi Van Dang (Reviewer #2).

The reviewers have discussed the reviews with one another and the Reviewing Editor has drafted this decision to help you prepare a revised submission.

The new feature of this work is the hypothesis that low cellular pH increases mitochondrial ATP through an enhanced mitochondrial proton gradient. The remaining findings (effects of low pH, targets TXNIP gene, nuclear translocation) have been previously reported from the Ayer laboratory. While the data presented seem to support that low intracellular pH elevates mitochondrial ATP, the mechanism by which this occurs is left largely undefined and explained only through inferences from known mitochondrial mechanisms, absent any data. A key conceptual issue that the authors did not address concerns the elevation of cellular ATP (Figure 5C for example reflects total cellular ATP increase by 4-fold while the "mitochondrial ATP sensor" increased by 20% in Figure 5D) and the insights from two recent papers. First, Walton et al. (2018) reported that low intracellular pH profoundly and rapidly inhibited mTORC1 activity through prompt dispersion of lysosomes. Secondly, Zheng et al. (*eLife,* 2016) reported a significant increase in ATP levels in neuronal cells with defective mitochondrial function in response to mTOR inhibition. Hence, the increase in ATP levels seen with low intracellular pH could be significantly due to decreased mTOR activity. Given this caveat, and the discrepancy between changes in mtATP vs. total cellular ATP (Figure 5C vs. Figure 5D) in response to low pH, an important question is whether the central model proposed by the authors is correct, particularly since the effect of low pH on mTOR activity was not considered. siRNA experiments are fraught with off-target effects and it is unclear how many siRNAs were used and how reproducible the results were for knocking down ATP5l, which disable mitochondrial ATP production. While Figure 5 illustrates the reduction of mtATP with siATP5l, total cellular ATP levels in response to low pH was not shown (as performed in Figure 5C).

A different question concerns the transcription mechanism of this new regulatory network. We don't have a clear understanding of how MondoA translocation to the nucleus is governed nor clear evidence of how or if MondoA acts directly at promoters. Are there ChIP-seq data detailing/investigating how these responsive promoters are targeted and activated? The authors suggest that MondoA acts directly only at TXNIP and one other gene, which would be highly unusual – why and how are these promoters selectively targeted?

Summary:

The manuscript by Wilde et al. reports the potential role of cellular acidosis in elevating mitochondrial ATP levels, which in turn increases the production of glucose-6-phosphate (G6P) through mitochondria-bound hexokinase 2 (HK2), and G6P is sensed by mitochondria-bound MondoA. G6P-activated MondoA then translocates to the nucleus to induce the expression of TXNIP. The transcription network used by MondoA to control glucose flux is very important but has not been well-defined mechanistically, and the current paper provides clear and compelling data regarding the effects of acidosis on mitochondrial ATP production, identification of genomewide targets, and the role for H2K, constitute a significant advance.

Essential revisions:

Three areas need to be addressed in a revised manuscript:

1) Address whether the effects observed could be due to inhibition of mTOR (i.e. determine the state of mTOR kinase activity and 4E-BP1 and/or S6K substrate phosphorylation under the experimental conditions used for the other studies presented in this paper, and assess whether the same effects on MondoA would be seen with an mTOR inhibitor).

2) Address the issue of siRNAs, were multiple ones used for a given protein, and/or would siRNA to TXNIP give the same result as knockdown of MondoA.

3) Provide a clearer discussion or more details of the transcriptional mechanism involved in targeting MondoA to the nucleus and to target genes, in particular address how MondoA targets are chosen and why it is considered that only a few genes are directly regulated. With these changes, the paper would clearly make an exciting and important contribution to the field.

*Reviewer #1:*

The transcription network used by MondoA to control glucose flux is important but poorly defined, and this manuscript contributes to our understanding by showing that: 1) cellular stress response to acidosis increases mitochondrial ATP production, which leads to elevated G6P levels and nuclear translocation of MondoA; 2) genomewide target gene identification; 3) demonstration by a variety of approaches that HK2 at the outer mitochondrial membrane is required for this regulation. Overall the data presented are clear and convincing and add important information to a novel regulatory network that has been a focus of this lab in recent years.

The weakness of the paper lies in the lack of transcriptional insights to the mechanism. We don't have a clear understanding of how MondoA translocation to the nucleus is governed, and most importantly, we do not have clear evidence of how or if MondoA acts directly at promoters. The data should be complemented with ChIP-seq data detailing/investigating how these responsive promoters are targeted and activated. The authors suggest that MondoA acts directly only at TXNIP and one other gene, but what is the evidence that they do this, and how do they work at those promoters – how are the other targets then identified? The data in Figures 1 and 2 are, in my view, background information that establish the conditions for the experiments and could be readily moved to supplement to make room for data addressing the mechanism of the transcriptional response.

*Reviewer #2:*

The manuscript by Wilde et al. reports the potential role of cellular acidosis in elevating mitochondrial ATP levels that in turn increases the production of glucose-6-phosphate (G6P) through mitochondria-bound hexokinase 2, and G6P is sensed by mitochondria-bound MondoA. G6P-activated MondoA then translocates to the nucleus to induce the expression of TXNIP. The new feature of this work is the hypothesis that low cellular pH increases mitochondrial ATP through enhanced mitochondrial proton gradient. The remaining findings have been previously reported from the Ayer laboratory regarding the sensing of G6P by MondoA, which was previously reported to be induced by low pH, and in turn MondoA induces its target TXNIP as well as other targets. While the data presented seem to support that low intracellular pH elevates mitochondrial ATP, the mechanism by which this occurs is left largely undefined and explained only through inferences from known mitochondrial mechanisms, absent any data. A key conceptual issue that the authors did not addressed regarding the elevation of cellular ATP (Figure 5C for example reflects total cellular ATP increase by 4-fold while the "mitochondrial ATP sensor" increased by 20% in Figure 5D) are the findings from two recent papers. First, Walton et al. (2018) reported that low intracellular pH profoundly and rapidly inhibited mTORC1 activity through prompt dispersion of lysosomes. Secondly, Zheng et al. (*eLife*, 2016) reported a significant increase in ATP levels in neuronal cells with defective mitochondrial function in response to mTOR inhibition. Hence, the increase in ATP levels seen with low intracellular pH could be significantly due to decreased mTOR activity. Given this caveat and the discrepancy between changes in mtATP vs total cellular ATP (Figure 5C vs. Figure 5D) in response to low pH raises the key question of whether the central model proposed by the authors is correct, particularly since the effect of low pH on mTOR activity was not accounted. siRNA experiments are fraught with off-target effects and it is unclear how many siRNAs were used and how reproducible the results were for knocking down ATP5l, which disable mitochondrial ATP production. While Figure 5 illustrates the reduction of mtATP with siATP5l, total cellular ATP levels in response to low pH was not shown (as performed in Figure 5C).

---

## [Author Response]

Essential revisions:Three areas need to be addressed in a revised manuscript:1) Address whether the effects observed could be due to inhibition of mTOR (i.e. determine the state of mTOR kinase activity and 4E-BP1 and/or S6K substrate phosphorylation under the experimental conditions used for the other studies presented in this paper, and assess whether the same effects on MondoA would be seen with an mTOR inhibitor).

We have directly tested whether treatment with DMEM^Acidic^ (~ pH 6.7) is sufficient to suppress mTORC1 activity and increase ATP levels leading to a stimulation of MondoA transcriptional activity. We show that DMEM^Acidic^ does not substantially reduce mTORC1 activity as measured by phosphorylation of S6, even in cells treated for 8 hours in the low pH medium (Figure 3—figure supplement 1). Note that all of our experiments were performed for 4 hours or less. The paper from Walton et al. used lower pH (6.3) than we did and for longer times (8 hour treatments), perhaps accounting for the difference in experimental outcome. Further the paper from Zheng et al. does show an increase in ATP levels following almost complete mTORC1 inhibition with rapamycin. They attributed this increase in ATP to a blockade in translation, rather than an increase in mitochondrial ATP synthesis. This finding is entirely inconsistent with our data showing that MondoA transcriptional activity is strongly dependent on mitochondrial ATP synthesis. Together these data suggest the DMEM^Acidic^ does not increase MondoA transcriptional activity by inhibition of mTORC1 and its secondary effect on cytoplasmic ATP pools.

2) Address the issue of siRNAs, were multiple ones used for a given protein, and/or would siRNA to TXNIP give the same result as knockdown of MondoA.

It is striking that TXNIP and its paralog ARRDC4 are the only two genes that show strong dependence on MondoA under both baseline and induced conditions (Figure 6). To begin to address the mechanism that underlies this phenomenon, we performed a MondoA ChIP-seq experiment from cells grown in DMEM or DMEM^Acidic^. We note that this will be the first ChIP-seq analysis for MondoA published in any cell system, which given the role of MondoA in glucose-dependent transcription is significant on its own. Our ChIP seq revealed that TXNIP and ARRDC4 are the two most highly occupied MondoA targets and not just by a little. Depending on the targets examined and the assay employed, MondoA occupies the TXNIP and ARRDC4 promoters between ~30 and ~200 fold more strongly than other targets. We provide a completely new Figure 7 with this data. This elevated binding likely explains why TXNIP and ARRDC4 expression is so highly dependent on MondoA. We note that for the next ~100 MondoA targets that rank after TXNIP and ARRDC4, only 4 are regulated more than 1.5 fold in MondoA knockout cells. Compared to ChIP studies examining occupancy of other transcription factors, MondoA binding to these sites is “strong”. We conclude that MondoA binding to this group of targets is either non-productive in terms of transcriptional activation or that these genes are only weakly dependent on MondoA. Our analysis suggests that high “affinity” MondoA binding to TXNIP and ARRDC4 is not related to a larger number of E-boxes in their promoters or whether their promoters lie in open chromatin regions. Thus, the phenomenon is clear but given the open ended nature of the experiments required to reveal the underlying mechanism(s), we believe determining the molecular basis responsible for increased MondoA binding to TXNIP and ARRDC4 is beyond the scope of this manuscript.

3) Provide a clearer discussion or more details of the transcriptional mechanism involved in targeting MondoA to the nucleus and to target genes, in particular address how MondoA targets are chosen and why it is considered that only a few genes are directly regulated. With these changes, the paper would clearly make an exciting and important contribution to the field.

The reviewers were concerned about the specificity of our knockdown experiments. We have clarified several points in the manuscript to address this concern. First, for our experiments examining MondoA function, we have used genetic or CRISPR MondoA knockout cells and used complementation in some cases to further implicate MondoA function. For the targets we knocked down using siRNA approaches, we used pools of 4 siRNAs against each target (ANT2, HK2 and ATP5I) and compared their activity to a si-Scrambled control. We also validated knockdown using either western blotting or RT-qPCR.

For HK2 and ATP5I, we used a number of orthogonal approaches to extend and confirm our findings. The involvement of ATP synthase (ATP5I knockdown) was implicated pharmacologically with oligomycin (Figure 2B). The experiments using 143Brho:ATP6/ATP8 cells implicate ATP synthase genetically (Figure 2G). The experiments with metformin and FCCP, implicate the contribution of a functional ETC to MondoA transcriptional activity (Figure 2A and B). In data we elected not to present, we used 143Brho cells complemented with mitochondria with mutations in cytochrome B (complex 3) to demonstrate the ETC requirement for lies downstream of complex 3, and therefore upstream of ATP synthase.

We also used multiple approaches to implicate the involvement of mitochondria-bound HK2. The requirement of a VDAC-HK2 interaction was tested with a mutant VDAC that can’t interact with HK2 (Figure 5A) and this data was confirmed with the bimolecular fluorescence complementation experiment (Figure 5B and C). The HK2 knockdown experiment is consistent with our previous experiments (Stoltzman, 2008), and was validated by showing that overexpression of wildtype HK2, but not kinase dead HK2 stimulated DMEM^Acidic^ induction of TXNIP. In data we elected not to show methyl-jasmonate, which is reported to remove HK2 from the surface of mitochondria blocked TXNIP expression following DMEM^Acidic^ treatment.

We have not validated the contribution of ANT2 by other approaches, but as above we used a pool of 4 different siRNAs for its knockdown and used immunoblotting to validate its knockdown. Further, the role on ANT is transporting mtATP from the matrix via VDAC to HK2 is consistent with the literature.